

# Wind-driven transport of fresh shelf water into the Labrador Sea Basin

Lena M. Schulze[1] and Eleanor Frajka-Williams[2]

[1]Geophysical Fluid Dynamics Institute, Florida State University, 77 Chieftain Way, 018 Keen Building, Tallahassee, USA
[2]University of Southampton, National Oceanography Center, Southampton, UK

**Correspondence:** Lena M. Schulze (lschulze@fsu.edu)

**Abstract.** The Labrador Sea is one of a small number of deep convection sites in the North Atlantic, that contribute to the meridional overturning circulation. Buoyancy is lost from surface waters during winter, allowing the formation of dense deep water. In the recent decades, mass loss from the Greenland ice sheet has accelerated, releasing freshwater into the high latitude North Atlantic. This and the enhanced Arctic freshwater export in the recent years have the potential to add buoyancy to the surface waters, slowing or suppressing convection in the Labrador Sea. However, the impact of freshwater on convection is dependent on whether or not it can escape the shallow, topographically-trapped boundary currents around Greenland and Labrador. Previous studies have estimated the transport of freshwater into the central Labrador Sea by focusing on the role of eddies. Here we use a Lagrangian approach, tracking particles in a global, eddy-permitting (1/12°) ocean model, to examine where and when freshwater enters the Labrador Sea basin in the surface 30 m. We find that most freshwater enters in the east (near the west coast of Greenland), consistent with previous expectations. Seasonally two peaks of freshening are observed. The first peak occurs in the spring and results from a large number of shelf water particles. The second peak, occurring in the fall, is due to the low salinity of the West Greenland current at this time of the year. We find that in these simulations surface wind-driven Ekman transport, rather than eddies, are responsible for the larger year-to-year variability in freshwater transport from the shelves to the central Labrador Sea.

## 1 Introduction

In the Labrador Sea, intense winter heat loss removes surface buoyancy, allowing deep mixing and the formation of deep dense water. The so-formed Labrador Sea Water (LSW) joins the deep western boundary current (DWBC) and is transported south as part of the Atlantic meridional overturning circulation (AMOC). Overall, the upper Labrador Sea is characterized by relatively salty Atlantic water offshore and cold, fresh water in the boundary currents over the shelves. In the salty basin, less cooling is required to cause static instabilities in winter that result in convection, making the Labrador Sea one of the prime regions for deep convection.

Freshening of the Labrador Sea surface water, in combination with air-sea fluxes, could reduce or eliminate convection due to the increase in surface buoyancy. In fact, freshening periods of varying intensity are not uncommon in the Labrador Sea (e.g. Houghton and Visbeck (2002)) due to its proximity to the fresh Arctic outflow as well as melt from the Greenland ice sheet.



An example of a complete shutdown of deep water formation due to additional surface buoyancy and weak air-sea fluxes was observed during the Great Salinity Anomaly (GSA) in the 1970s (Dickson et al., 1988; Gelderloos et al., 2012). Convection later resumed due to a strong air-sea fluxes as well as advection of saltier water (Gelderloos et al., 2012). Increased freshwater input in the North Atlantic, as observed in the last decades (Bamber et al., 2012), may increase the amount of freshwater

entering the Labrador Sea while decreasing the deep water formation rate. Model simulations indicate that predicted rates of freshening in the region will cause a 20% change in the strength of AMOC (Häkkinen, 1999; Manabe and Stouffer, 1995; Jahn and Holland, 2013; Robson et al., 2014). Until 2005, increased freshwater was not detectable as a persistent freshening signal in the upper Labrador Sea (Yashayaev, 2007). However, more recent studies, using ocean observations such as Argo floats and ship-based hydrography, show that the surface layer of the North Atlantic, including the Labrador Sea, has freshened. Simulta-

neously, deep densities have decreased over the last decade which has possible impacts for the Atlantic overturning circulation (Yashayaev et al., 2015; Robson et al., 2014; **?**). Despite the reduction in salinity, recent years (2014 – 2016) showed the return of deep convection and the formation of a new LSW class (Yashayaev and Loder, 2016).

Early, so-called 'hosing experiments', were performed in coarse resolution numerical models to simulate large amounts of freshwater released during paleoclimate events. These simulations showed that freshwater added to the Arctic spread uniformly

across the entire North Atlantic, including the Labrador Sea (e.g. Weaver et al. (1994)). Higher resolution models suggest however, that additional freshwater in the Labrador Sea may be confined to the shelf region (Myers, 2005) where it would have less influence on the properties of convection region. In a comparison between three different models (with resolutions of $1/2^\circ$, $1/4^\circ$ and $1/12^\circ$), an increase of fresh melt water from Greenland was found in the central Labrador Sea in all models Dukhovskoy et al. (2015). The freshwater entered mainly from Baffin Bay and the south, but the amount of freshwater that reaches the region

of convection differs between the models. Additionally, the study suggests that any freshwater signal reaching the Labrador Sea would be obscured by the increased salinity of the Atlantic Water entering the region at the same time (Dukhovskoy et al., 2015).

On seasonal timescales, freshwater is observed to enters the basin in a small pulse in the spring and a second, larger pulse in the fall (Schmidt and Send, 2007). These authors attribute the freshwater of the first peak to the Labrador Current and the

second, larger peak to the West Greenland Current. This is consistent with Lilly et al. (2003) who also identify the West Greenland Current as the primary source of the freshening in the Labrador Sea. Additional freshwater joins the Labrador Current from Davis Strait and Hudson Strait. Evidence has been brought forward pointing to instabilities in the Labrador Current that could lead to advection of freshwater into the basin (LeBlond, 1982; Cooke et al., 2014). In fact, Cooke et al. (2014) argue that this indicates a direct connection between the Labrador Current and central basin salinities. Such a connection would further

support the idea of a Labrador Current source to the fall freshening in the central Labrador Sea.

In the past, studies have concentrated on eddies as the main mechanism by which heat and freshwater are imported into the basin. Eddies originating at the boundary current can carry warm and buoyant water (Lilly et al., 2003; Jong et al., 2014) and have been associated with seasonal freshening (Chanut et al., 2008; Katsman et al., 2004). Eddies with a core of Irminger Sea Water, termed Irminger Rings, are shed from the boundary current near the northeast corner of the basin (around 64°N, 54°W).

When assuming that 30 eddies are shed from the boundary current each year (as observed by Lilly et al. (2003)), up to 50 – 80



% of the wintertime heat loss to the atmosphere can be balanced by eddies advecting heat (Lilly et al., 2003; Katsman et al., 2004). However, eddy advection can only account for $\sim 50\%$ of the freshwater that is needed to explain the seasonal freshening in the basin (Lilly et al., 2003; Hatun et al., 2007; Straneo, 2001). Hence, there is an unresolved discrepancy between the advection of freshwater by eddies and that required to explain the annual freshwater gain in the basin. Observational studies may underestimate the number of eddies due to the coarse resolution of altimetry data relative to eddy size. Additionally, other dynamics might also be important in allowing freshwater to enter the basin. Here, we consider whether or not surface Ekman transport may be important in this process.

Every year, substantial buoyancy is lost from the Labrador Sea basin during the wintertime convection. This buoyancy is replenished by surface heat fluxes and lateral buoyancy fluxes (Straneo, 2001), that have both a time-varying and a mean component. Here we focus on these aspects using a numerical model to better understand the changing processes involved in the freshwater fluxes into the Labrador Sea. This includes time-varying eddy fluxes and wind-driven Ekman fluxes.

In this study we will use Lagrangian trajectories in a high resolution (1/12º) numerical model to investigate how, when, and where surface freshwater from boundary currents enters the central Labrador Sea. In particular, the relative importance of eddies versus wind in allowing freshwater to escape the shelves and enter the basin. In Section 2, we describe the model and methods. In Section 3, we outline the typical pattern of shelf-edge crossings, their salinity and origin. In Section 4, we consider the variability of crossings and its relationship to eddy and wind-activity in the region. We conclude in Section 5 and 6 with a summary and discussion.

## 2 Data and Methods

We use output from a 1/12º numerical model to compute offline Lagrangian trajectories of water particles to better understand where and how water crosses into the central Labrador Sea. In the following, we describe the numerical model and compare velocity and hydrography to observations (Section 2.1). We then give an overview of the particle-tracking software (ARIANE) and detail particle releases (Section 2.2), as well as explain the criteria for a 'crossing' from shelf-to-basin in the Labrador Sea (Section 2.3). Since a large part of this work will focus on where these particles originate, we define the possible regions of origin in Section 2.4. Trajectories are ideally suited to identify the pathway and origins of water parcels with associated temperatures and salinities. These are key to our focus on processes driving the movement of water from the shelves to the central basin.

### 2.1 NEMO data

For this study, output from the high-resolution global ocean circulation model NEMO ORCA V3.6 ORCA0083-N06 (Nucleus for European Model of the Ocean, NEMO N06 from here on) is utilized (Madec, 2008; Marzocchi et al., 2015; Moat et al., 2016). The model has a horizontal resolution of 1/12º with a tri-polar grid (with one pole in Canada, one in Russia and one on the South Pole) to avoid numerical instability associated with convergence of the meridians at the geographic North Pole. Resolution is coarsest at the equator (9.26 km) and increases to about 4 km in the Labrador Sea. This allows the model to





resolve some mesoscale eddies. Smaller features are parameterized.

The model has 75 vertical levels that are finer near the surface (about 1 m) and increase to 250 m at the bottom. The bottom topography is derived from the 1-minute resolution ETOPO bathymetry field of the National Geophysical Data Center (available at *http://www.ngdc.noaa.gov/mgg/global/global.hmtl*) and is merged with satellite-based bathymetry. Model output

is produced every 5 days. Lateral mixing varies horizontally according to a Bi-Laplacian operator with a horizontal eddy viscosity of 500 m$^4$/s. Vertical mixing at sub-grid scales was parameterized using a turbulent kinetic energy closure model (Madec, 2008). Background vertical eddy viscosity and diffusivity are $10^{-4}$ m$^2$/s and $10^{-5}$ m$^2$/s, respectively.

The model is forced by the Drakkar Surface Forcing data set V5.2. developed by the DRAKKAR consortium (*http://www.drakkar-ocean.eu/*) supplying air temperature, winds, humidity, surface radiative heat fluxes and precipitation. It is integrated for the

period 1958 – 2012, with a horizontal resolution of 1.125º (Dussin et al., 2014; Brodeau et al., 2010). Precipitation, downward shortwave and longwave radiation are taken from the CORE forcing data set (Large and Yeager, 2004) while wind, air humidity, and air temperature are derived from the ERA-Interim reanalysis fields. Surface momentum in the model is applied directly as a wind stress vector using daily mean wind stress. To prevent unrealistic salinity drifts in the model due to deficiencies in the freshwater forcing, the sea surface freshwater fluxes are relaxed toward climatologies by 33.3 mm/day/psu, corresponding

to a relaxation timescale of 365 days. The subsequent analysis does not attempt to calculate any freshwater budgets or compare model salinities to observations. Instead we focus on pathways of fresh versus salty water into the basin as well as month-to-month and interannual changes in the freshwater that is transported to the basin within the model.

The sea ice module used is from the Louvain-la-Neuve sea ice model (LIM2), (Timmerman et al., 2005). No-slip conditions are implemented at the lateral boundaries - except in the Labrador Sea where a region of partial slip is applied. This is done to

favor the break up of the West Greenland Current into eddies (as observations have suggested). For each model cell, the model uses the ice fraction to compute the ice-ocean fluxes combined with the air-sea fluxes to provide the surface ocean fluxes. No icebergs are implemented in this version. The absence of icebergs in our study is discussed in Section 5.

In the model, the ocean is bounded by complex coastlines, bottom topography and an air-sea interface at the surface. The major flux between the continental margins and the ocean is a mass exchange of freshwater through river runoff, modifying the

surface salinity. There are no fluxes of heat and salt across solid boundaries between solid earth and ocean, but the ocean exchanges momentum with the earth through frictional processes. Initial conditions for the model were taken from Levitus et al. (1998) with the exception of high latitudes and Mediterranean regions. Here PHC2.1 (Steele et al., 2001) and MEDATLAS (Jourdan et al., 1998) are used, respectively. The model is run for the period of 1958 – 2012. Here we analyze the time period of 1990 – 2009, for which eddies and wave fields (Rossby waves) had ample time to spin up.

### 2.1.1 Model evaluation

We evaluate the model in terms of it being an acceptable tool to our scientific question. The NEMO simulation used in a variety of study involving the North Atlantic. For example, Moat et al. (2016) found that the model well represents the variability of heat transport at 26.5 ºN. Substantial changes have been incorporated to improve the representation of boundary





currents, interannual variability and depth of mixed layers compared to previous 1° and 1/4° runs. Changes implemented in the model were: 1) the wind forcing was made more consistent reaching back to 1958 (more information at *www.drakkar-ocean.eu/forcing-the-ocean/the-making-of-the-drakkar-forcing-set-dfs5*), 2) changes in topography and 4) the use of a partial

slip condition along western Greenland (Quartly et al., 2013). Together with the changes in topography, the partial slip condition promotes the formation of eddies in this region and results in an improved pattern of salinity field and velocities (**Figure 1**).

The deepest winter mixed layer in the Labrador Sea basin seen in the N06 model are located in the western basin, consistent with observations (Pickart et al., 2002; Våge et al., 2008; Schulze et al., 2016) (**Figure 1**). The correct location and magnitude

of the mixed layers shows that NEMO N06 well represents the boundary currents and advection of freshwater and heat into the basin. Without this representation, the basin stratification would be weaker and mixing would be stronger, resulting in mixed layers that are much deeper then in the observations.

The mean NEMO N06 surface salinities in the Labrador Sea are shown in **Figure 1** together with data from Argo floats in the region (see *www.argo.com* for information about this data). Argo data are generally not available on the shelves where

water is shallower than 1000 m (with some exceptions) but the deep basin properties are reasonably well observed. Both the surface salinities in NEMO and from Argo data show freshest water (below 34.8) in the coastal regions. At Cape Farewell (southern tip of Greenland), salinities are high, $\sim 34.9$ in NEMO and above 34.99 in the Argo data. The salinity of the basin is $\sim 34.85$ in NEMO with a saltier region in the northwest (34.875 - 34.9) and a fresher region in the northeast (34.8 - 34.5). A similar salinity distribution can also be found in the Argo data. The saltiest region is in the western basin with salinities

around 34.9. The freshwater in the northeast extents further into the basin but with salinities around 34.5 - 34.8. While there are some differences, both, the model and observations show increased salinities in the western Labrador Sea as well as a band of slightly lower salinities extending across the Labrador Sea. This band joins the high salinities in the southeastern Labrador Sea. Seasonal cycles of the basin-averaged salinities in NEMO and from Argo data are in phase with peak salinities in February – March and the freshes water in September. Modeled salinities are overestimated by around 0.1 between November - June.

The NEMO N06 model shows a strong WGC and Labrador Current, as well as flow from Baffin Bay and Hudson Strait (**Figure 1**). The region around 62°N and 52°W, described as the region of high EKE (Eddy kinetic energy) in many studies, is characterized in NEMO N06 by an energetic WGC and the formation of eddies. Along the coast of the Labrador peninsula, the flow is separated into two currents; a coastal flow, and the main branch of the Labrador Current. The coastal current is mainly fed by outflow from Hudson Strait and is separated from the Labrador Sea Basin (Han et al., 2008). The flux between the

Labrador Sea and Baffin Bay experiences a strong seasonal cycle in NEMO that is consistent with hydrographic observations in this region (Myers, 2005; Curry et al., 2014; Rykova et al., 2015).

Along the east coast of Greenland, the EGC is also split into a coastal branch and the main branch. Such coastal flow has also been observed in the past by e.g. Sutherland and Pickart (2008). Luo et al. (2016) show a similar flow pattern in their model study, with current speeds of up to 1 m/s in the WGC and LC. However, their data show very little eddy activity in the

northeast, and a strong outflow from Baffin Bay, but no outflow from Hudson Strait. It is harder to compare the velocities to the higher resolution model used in Böning et al. (2016), since in their model, ice covers Baffin Bay, Hudson Strait and much





of the Labrador Shelf. However, they also see a strong and steady WGC that becomes unstable around 62°N and 52°W.

The region of high EKE in the northeast corner of the Labrador Sea basin has been described in many studies. Using merged along-track TOPEX/Poseidon and ERS data for 1997–2001, Brandt et al. (2004) find the region of largest EKE in the WGC at

$\sim$ 62°N, in water shallower than 2500 m. Maximum values are found to be as high as 700 cm$^2$/s$^2$. This differs from the gridded AVISO data, where the maximum EKE is located further offshore and does not exceed 100 cm$^2$/s$^2$. Their EKE reaches values of $\sim$ 300 cm$^2$/s$^2$ inside the basin close to the northeast corner, consistent also with Chanut et al. (2008), Katsman et al. (2004), and Lilly et al. (2003). The EKE calculated from the NEMO data has very similar values with the maximum EKE in the same locations shown by Brandt et al. (2004). In particular, the region of the highest EKE is located outside the 2500 m isobath at

around 62°N, with values that reach 600 cm$^2$/s$^2$. Inside the basin, the northeast is characterized by EKE values of up to 200 cm$^2$/s$^2$. The highest values of EKE in the model used by Luo et al. (2016) are consistent with the location of the highest EKE values in NEMO. Altimetry data on the other hand, does not have elevated EKE inside the basin (Brandt et al., 2004). Brandt et al. (2004) further observe that the EKE in the WGC is on average more than 300 cm$^2$/s$^2$ higher than in the central LS, and that the minimum/maximum EKE in the WGC and the basin occurs in September/January. Both are also true for the NEMO

data with EKE timing and values that compare well to satellite data.

## 2.2 ARIANE and experiment setup

The off-line Lagrangian tool ARIANE is used to track particles using velocity fields output from the NEMO model. ARIANE is available at *http://www.univ-brest.fr/lpo/ariane* and described in detail by Blanke and Raynaud (1997) and Blanke et al. (1999). For each 5 day timestep of the model the trajectories are analytically solved, respecting the mass conservation of the

model within each grid cell.

For this study, particles were released every 10 days over the period of 1990 – 2009 at 264 points in the Labrador Sea basin (**Figure 2**). Most freshwater is contained in the upper 30 m and we place our release points were place at three different depths (0 m, 15 m, and 30 m). This results in 28,512 particle releases each year, for a total of 570,240 particles over the 20 year period of 1990 – 2009. These particles provide a statistical description of water pathways in the Labrador Sea. Each particle is tracked

backwards for one year.

## 2.3 Particles crossing into the basin

We refer to the Labrador Sea basin as the region that is offshore of the 2500 m isobath. This region is encircled by the boundary currents which are usually centered at this isobath (**Figure 1c**). A particle is considered to have entered the basin if it crossed the 2500 m isobath from shallow into deeper water within the top 30 m of the water column. If a particle crosses the isobath

multiple times, only the last time before reaching its release point (integrated backwards in time) is considered. In addition, the particle has to move at least 50 km away from the 2500 m isobath to be considered as within the basin. This criteria ensures that the particle has left the boundary current completely. The 50 km threshold was determined by averaging the velocities of the basin as a function of distance from the 2500 m isobath (not shown). Average velocities exceed 0.25 m/s within 20 km of the 2500 m isobath but decrease to 0.1 m/s at a distance of 50 km. There is little to no influence of the boundary currents





beyond this distance and velocities remain constant at 0.1 m/s.

Note, particles are only considered in this study if they crossed into the basin within the top 30 m. Between 1990 – 2009, a total of 570,240 particles were released of which 230,147 (40%) entered the basin within the top 30 m during their lifetime of one year (**Table 1**, second line: Crossing <30 m). Additionally, we will consider crossings that occur within 7 month of the particle release (i.e., particles that crossed from the shelves to the Labrador basin within the 7 month prior to when they were initialized in the central Labrador Sea). This is the case for a total of 205,929 particles. A randomly chosen ensemble of trajectories of particles in this category is shown in **Figure 3**. The 7-month cut-off allows the seasonal cycle to be resolved, but the results presented below are not strongly sensitive to the choice of a cut-off time. Of the remaining 323,084 trajectories that are not categorized as crossings according to the above criteria, 1657 crossed below 30 m and 15,352 were initialized in the basin and remained there during their one year lifetime (**Table 1**). The largest number of particles (56%) enters the basin from the south but never crosses the 2500 m isobath.

## 2.4 Regions and Water Sources

The 2500 m isobath, which we consider to be the boundary between shelf and basin, is split into three areas: Southeast, Northeast and West (**Figure 2**). Particles crossing into the basin via three sections is traced to its source. We consider five sources: Hudson Strait, Baffin Bay, East Greenland Current (EGC) inshore, and EGC offshore and water from other sources in the North Atlantic (also referred to as North Atlantic water), (**Figure 2**). The EGC inshore and offshore sources at the east Greenland coast are separated by the 1000 m isobath. This isobath coincides with a strong surface salinity gradient of 0.6 psu between the fresh inshore water and saltier offshore water (not shown). If a particle passed through either the EGC inshore or offshore regions at any point during its lifetime it is considered to have its origin in the EGC. A particle is considered to originate from Hudson Bay if at any point it was located west of 65°W. Similarly, every particle that passed through the region west of Greenland and north of 65°N has its origin in the Baffin Bay. All other particles must originate elsewhere and are of North Atlantic origin.

The majority of the particles entering the Labrador Sea basin (80%) originate in the EGC (both, inshore and offshore portions of the current, **Figure 2**). Specifically, 95,810 (46.5%) of the 205,929 particles originated in the offshore section of the EGC; 69,028 (33.5%) originate in the inshore EGC (hence from the shelf). A much smaller number (29,406 or 14%) entered the Labrador Sea basin from elsewhere in the North Atlantic. During the 20 years considered here, only 153 particles (≪ 1%) originated in Baffin Bay and four in Hudson Bay. Because of this small number (compared to the number of crossings from the other sources), Baffin Bay and Hudson Bay will not be considered in the results from here on. Due to the one-year lifetime of the particles, 5.5% (11,528) of particles that crossed into the basin did not originate in one of these five regions. Hence, at the end of their lifetime they were located outside the basin but had not left the Labrador Sea.



## 2.5 Probability of crossings

Below we will present the number of crossings as a probability of particles enter the basin in a certain region and during a specific time period (e.g. monthly or yearly). The probability is calculated by dividing the number of crossings in a certain region or within a certain time period (monthly or yearly) by the total number of crossings.

## 2.6 Ekman Transport

To calculate the expected Ekman transport for a homogeneous ocean into the basin we use the ERA-Interim reanalysis 10-meter wind product for 1990 – 2009. Daily winds are interpolated onto the southeast, northeast and west sections (**Figure 2**) and the along and across velocity components projected onto the respective section to be along ($\tau_\parallel$) and across the section ($\tau_\perp$). In this way, the Ekman transport across the section is given by

$$V_{\perp,ek} = \frac{\tau_\parallel}{f\rho} \tag{1}$$

where $\tau$ is the mean wind stress along the section, (calculated following Large and Pond (1980)), $f$ the Coriolis force, and $\rho$ the mean water density.

## 2.7 Error Analysis

Errors on the number of crossings and salinity are calculated using a Monte-Carlo approach. For the calculation of the error, a 90% subset of the variable (number of crossings and salinity) is selected randomly with replacement and the mean of the variable across the subset is calculated. The process is repeated 5000 times, after which the distribution of the estimated mean can be used to determine 95% confidence intervals. In this way, we can estimate how confident we are in the calculated mean of the variable. The error evaluates the robustness of our estimates using a reduced number of particles but does not address any uncertainties associated with model shortcomings in salinity or velocity fields.

## 3 Geography of Crossings

In this section, we discuss the geography of crossings identified by the ARIANE particles in the NEMO 1/12 º model run. In general, the highest probability of particles crossing into the basin occurs in the southeast and northeast of the Labrador Sea (**Figure 4**). In the west, the probability of crossings is about four times smaller compared to the east. It is worth noting, however, that the probability is slightly elevated south of 57ºN (section IV, and V in **Figure 5**). The southeast has the highest probability of particles entering the basin (section I, and II) with average salinities of 34.98. That is 0.04 psu higher than the average salinities of particles crossing in the northeast (34.94). Low salinity water crossing in the northeast (section II and III) combined with the high probability of crossings results in a large likelihood of freshwater entering the basin at these locations. Crossings in the southeast on the other hand do not supply any freshwater to the basin overall, due to the high salinities of the crossing particles here. Hence, the model output shows two distinct pathways of water into the basin; salty water enters in the southeast and freshwater in the northeast.





## 3.1 Crossings by water sources

To analyze the origin of the fresh and salty water that enters the basin in the north- and southeast we consider water originating in the EGC (inshore and offshore) as well as water from other regions in the North Atlantic separately. Water from the offshore EGC source is most likely to enter the basin in the southeast, a short distance downstream from Cape Farewell (**Figure 5**).

These particles are salty with an average of 34.97 psu. The main pathway of EGC inshore water into the basin is about 200 km further north along the boundary. Compared to the EGC offshore water, the water here is much fresher with salinities as low as 34.91 psu. Water with origin elsewhere in the North Atlantic primarily enters the basin a short distance from Cape Farewell, via the southeast (section I). The water is about 0.04 psu fresher than the EGC offshore water that also crosses the boundary primarily at this location. Further along the 2500 m isobath, the salinities of the water from all three sources are comparable

and the probability of crossings decreases to close to zero (section III – VI). For all three water sources, the speed at which particles cross into the basin is comparable (not shown).

In summary, the large amount of EGC offshore water crossing into the basin in the southeast results in an influx of relatively salty water to the basin. The EGC inshore water, on the other hand, enters farther north and brings much fresher water to the basin. Compared to the high probability that water enters along the eastern side of the basin, the crossings along the western side

are negligible. Additionally, in our study the contribution to freshwater fluxes from the water of other North Atlantic sources is small compared to the contributions of the inshore and offshore EGC water. Therefore, we focus on water originating in the EGC and entering the Labrador Sea basin along the eastern side.

## 4  Variability of crossings

In the following section, we identify the seasonal and interannual variability of particle crossings of the Lagrangian approach

in the 1/12 ° model run.

### 4.1  Seasonality of crossings

We divide the crossing particles according to their origin (EGC inshore or offshore) and the location at which they enter the basin (southeast or northeast) to investigate the seasonality of water entering the basin.

In the southeast, the probability of particles of EGC inshore and offshore origin to enter the basin is largest in March (**Figure**

**6**). However, the probability of EGC offshore water entering the basin is twice as high as the probability of inshore water crossing ($10.8\% \pm 0.2\%$ and $4.6\% \pm 0.1\%$, respectively). In addition to the high probabilities in March, there is also a high probability of inshore water crossing in January ($4.2\% \pm 0.1\%$). In summer the crossing probability is about half of the one in March for both inshore and offshore water. During the minimum in July, offshore water crosses with a likelihood of $3.8\% \pm 0.1\%$ and inshore water with a probability of $0.1\% \pm 0.02\%$.

In the northeast, the crossing probabilities of EGC offshore water is low in the northeast with probabilities varying from 1.3% in February to 3.2% in October. The seasonal cycle of the inshore crossings is similar (in timing and magnitude) to the





southeast region, with maximum probabilities in January and March and a minimum in the summer. Inshore water is about twice as likely as offshore water to enter during the time of convection in November – April (5% ± 0.2% versus 1.8% ± 0.1%, respectively). In the summer, the inshore crossings drop to almost zero while offshore water keeps entering the basin with probabilities of ∼3.5% ± 0.1%.

In the southeast, EGC inshore and offshore water entering the basin is saltier than 34.95 except during May and December. In the northeast, on the other hand, the EGC inshore water brings fresh water into the basin year-round with the exception of July, August, and November. In other words, the seasonal cycle of inshore water entering the basin in the northeast is characterized by two pulses of fresh water, one in December – April and a second, shorter pulse in September. The EGC offshore water also freshens during these two periods, but this freshening is much weaker and salinities remain close to the reference salinities.

### 4.1.1 Seasonal role of winds and eddies

Three-monthly composites of EKE and wind speeds show that the northeast portion of the Labrador Sea experiences high EKE in the spring and weak EKE in the fall. Winds are predominantly northwesterly. (**Figure 7**). Northwesterly winds will result in a southwestward Ekman transport which, for the Greenland side of the Labrador Sea, will be in the offshore direction. This effect is largest in the winter, followed by the spring, with nearly zero average transport in the summer.

There is only weak seasonality of EKE near the southeast section with values around 80 cm$^2$/s$^2$ all year (**Figure 8**). In the northeast, on the other hand, EKE values are much higher with an average of nearly 300 cm$^2$/s$^2$ and an amplitude of ∼ 200 cm$^2$/s$^2$. The maximum EKE is observed in February – March. Ekman transport into the basin is strongest in the southeast with peak values of around 4 mSv in March and a minimum of -1 mSv (transport out of the basin) in June. (Note that this is the overall water transport due to the winds, not the freshwater transport.)

In the southeast, the peak of EGC inshore and offshore crossings coincides with the peak of the Ekman transport. In the northeast, on the other hand, the peak of EKE and Ekman transport coincides only with the peak of inshore crossings.

Due to the similar timing of the seasonal maxima in EKE and winds, we cannot use the seasonal cycles to distinguish between their potential roles in transporting water from the shelves into the basin. In order to further separate their effects further, the interannual variability of the number of crossings, EKE and Ekman transport are evaluated below.

### 4.2 Interannual variability of crossings

The annual averages of the probability of crossings and their salinities are determined for the southeast and northeast sections (**Figure 9**). Throughout the entire period of study, offshore water is twice as likely to enter the basin via the southeast compared to inshore water. The inshore water crossings are relatively constant throughout the 20 year period, with no apparent long term trend. However, there seems to be a decrease in the amount of offshore water that enters the basin. In the northeast, the EGC inshore and offshore water have the same probability of entering the basin.

In both regions, the offshore water transports mainly salty water (relative to the reference salinity). The inshore water is relatively salty in the southeast and fresher in the northeast. The salinities during 1993 – 1995 are anomalous in both regions.





During these years the inshore water was much fresher along the entire eastern boundary than during other years. Other periods of elevated freshwater fluxes would have occurred in 1999, 2004, and 2007 – 2009 when salinities of the inshore water fell

below the reference salinity.

For all of the 20 years, the EGC offshore water is the main source of salty water entering in the southeast. Due to the low number of crossings, the EGC inshore water did not contribute significantly to fresh or salty water entering the basin in the southeast. In the northeast, where both sources are equally likely to enter the basin, EGC inshore water cause large freshwater fluxes in certain years (1993 – 1995, 1999, 2004, and 2007 – 2009), due to its much lower salinities.

**4.2.1   Interannual role of winds and eddies**

We now compare the interannual crossing probabilities to the anomalies of the Ekman transport and EKE. In particular, three-month averaged timeseries of EKE, Ekman transport, and probability of crossings in the southeast and northeast are constructed. To consider variations beyond the seasonal cycle, the mean seasonal cycle for 1990 – 2009 is removed and the resulting anomalies are shown in **Figure 10**, together with the crossing probabilities. The timeseries for EKE and Ekman trans-

port are correlated with the probability anomaly using the Pearson method (Thompson and Emery, 2014).

As mentioned above, previous studies have investigated eddies as the main mechanism through which water enters the basin from the shelf. Here, we find that anomalies of the crossing probabilities in the southeast are not significantly correlated with the EKE anomaly in this region. The crossing probabilities do, however, have a low but significant correlation with the Ekman transport ($r = 0.43$, **Table 2**). This relationship is more pronounced in the northeast, where the variability of the crossings is

highly correlated to the variability in the Ekman transport ($r = 0.73$). In other words, in the northeast the variability in Ekman transport can explain the majority of the variability in the crossing particles. In the NEMO model used here, EKE, and hence eddies, do not play a role in the variability of crossings (correlation of $r = 0.05$). One possible exception to this may be in the northeast, during the period 1998 – 2002, where there appears to be a period of transient correlation between crossing probability and EKE.

When repeating this calculation separately for the inshore and offshore crossing probabilities, only the probability of the inshore water crossing is significantly correlated to the Ekman transport (not shown). Furthermore, the correlation between EGC inshore water and Ekman transport is stronger in the northeast ($r = 0.72$), than the southeast ($r = 0.54$), though both are significant.

For a spatial view of the different conditions during times with high versus low crossings, maps of EKE and Ekman transport

and the mean salinity of the Labrador Sea are calculated (**Figure 11**). In particular, the maps are comprised of months when the probability of crossings in the southeast and northeast is outside of a two standard deviation envelope. At times when crossing probabilities are high, the EKE in the northeast is weak and the Ekman transport across the eastern side of the basin is stronger, compared to times with anomalously low crossings. Additionally, the surface salinities on the Greenland shelves and the central Labrador Sea basin are 0.2 psu fresher when the probability of crossings is high. The West Greenland Current at Cape Farewell is also fresher in this scenario.

The following pattern emerges: During times with anomalously high crossings, the EKE in the northeast, just onshore and

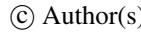



adjacent to the 2500 m isobath was on average 100 cm$^2$/s$^2$ lower than during the months with the fewest crossings. The north-east region just inside the 2500 m isobath, on the other hand, has similar values for both scenarios. Much larger differences are

found in the Ekman transport. During times of anomalously low transport, winds force water into the basin along the northern boundary, but the Ekman transport is parallel to the eastern boundary and hence results in weak cross-shelf Ekman transport here. This is accompanied by higher than average salinities on the shelves. When the number of crossings is high however, the Ekman transport is strong and perpendicular to the eastern boundary, allowing the water to spread away from the shelf region into the Labrador Sea basin. This leads to an overall freshening of the basin.

## 10   5   Discussion

We use the ocean model NEMO and the Lagrangian particle tracking tool ARIANE to assess the major routes and mechanisms of freshwater in the Labrador Sea basin that are important to understand how the freshwater released from the Greenland ice sheet or Arctic may influence the region of deep convection in the Labrador Sea. The model used here is 1/12$^o$ which is eddy-permitting but not eddy-resolving at these latitudes. By determining the relative likelihood and associated salinities, we can

evaluate the cause of freshwater changes in the basin. In addition, by investigating the temporal variability of the cross-shelf movements of water, we can determine likely forcing mechanisms of the cross-shelf transport. In particular, we have considered the role of Ekman transport and eddy fluxes (given by eddy kinetic energy) for the exchange between the boundary and basin in the upper 30 m.

Lagrangian trajectories suggest that in the NEMO model, 80% of the water entering the basin each year in the top 30 m originates in the East Greenland Current. It reaches the Labrador Sea via the West Greenland Current before crossing into the basin along the eastern boundary. In comparison, water originating from other regions such as Baffin Bay and Hudson Straight are negligible. While there are some possible shortcomings in how the circulation in these regions is represented in the model, our findings are consistent with previous studies that anticipate water entering the region in the east (Myers, 2005),

coincident with freshening near Greenland (Schmidt and Send, 2007) and near the location of high EKE (Lilly et al., 2003). Here, the dominant pathway of water particles from the boundary to the central basin is found to be in the northeast. There is a significant role of wind-driven transport which seems to force the interannual, and possibly the seasonal, variability of cross-shelf exchange in the model. These results show that Ekman transport may also play an important role in the cross-shelf transport, and offer some guidance on likely regions where the cross-shelf transport may occur. While the Hudson Strait and

Baffin Bay waters played little role in the freshwater transport in this model, due to their extremely low salinities, it would be worth verifying with observational data that there is no additional pathway for freshwater from these sources to the Labrador basin. In addition, higher resolution models might be able to resolve eddies in the Labrador Sea much better. This might be needed to really understand the role eddies play in transporting freshwater to the basin in this region.



Seasonally, the largest number of crossings is observed in the spring, but fluxes into the basin continue at a lower rate throughout the year. This is consistent with previous observationally-based estimates using a budget framework that also showed continuous fluxes of water into the basin (Straneo, 2001). Freshwater is advected into the basin in two pulses, in the spring and in the fall, as was also observed by previous studies (Schmidt and Send, 2007; Straneo, 2001). Due to the different methods applied in the studies (e.g. deeper surface layers and different reference salinities) and the saltier model used here, the absolute magnitudes of the freshening pulses are not explicitly compared. However, the results are consistent in the timing of the freshening and their relative magnitudes, with the second pulse about three times stronger than the first pulse.

One of the unique benefits of a Lagrangian approach is the ability to determine the statistical source of the water entering the basin. We investigate the origin of the freshwater that enters the basin, finding that the water originating in the inshore region of the East Greenland Current and entering the Labrador Sea in the northeast is responsible for the first (March – April) freshening pulse. This water alone is able to flux large amounts of freshwater into the basin. However, at the same time, large amounts of salty EGC offshore water enter the basin in the southeast. This counteracts and weakens the freshening in the spring. The large fall pulse (September – October), on the other hand, is the result of a combination of relatively low salinity water from the EGC offshore source and very fresh EGC inshore water. The two water masses enter the basin in two different regions, the EGC offshore water in the southeast and the EGC inshore water in the northeast.

Our results show that the interannual probability of freshwater entering the basin was highest in the mid-1990s, with other maxima in 1999, the early 2000s and mid-2000. Several other years stand out as well in terms of large probabilities of freshwater fluxes, such as 1999, 2003 –2004 and 2007 – 2008. The water responsible for these freshening periods originated in the inshore part of the EGC, while the EGC offshore water did not contribute. A freshening in the late-1990s was observed by Häkkinen (1999), with fresh anomalies located mainly on the shelves. This is consistent with the model where such a freshening period took place due to the fresh EGC inshore water.

Due to the remarkably high correlation between the Ekman transport and crossing probability, we suggest that wind forcing plays the primary role in the variability of freshwater transport near the surface, and in allowing fresh shelf water to enter the basin. This conclusion is consistent with model results presented by Luo et al. (2016). In summary: As water rounds Cape Farewell and enters the Labrador Sea a large amount of the offshore water crosses into the basin. The inshore water on the other hand spreads away from the coast, off the shelf and towards the basin, due to Ekman transport. The offshore water enters the basin due to other mechanisms (not addressed in this study) and hence the number of crossings of this water is not significantly correlated to the Ekman transport.

While the Lagrangian approach us to investigate into the timing, relative numbers of crossings and salinities of these crossings, they cannot be directly related to a net transport across a section. For a quick comparison, we calculate the freshwater fluxes due to Ekman transport directly from the model data by using wind and mean model salinities of the top 30 m across





eastern sections. This shows that Ekman transport is responsible for a mean inflow of 1.5 mSv of freshwater. To estimate eddy fluxes across the same sections, we consider $v = \overline{v} + v'$ where $v$ is the total volume flow, $\overline{v}$ the time-mean, and $v'$ a deviation from the time-mean and hence the volume flux due to eddy fluxes. This is done for the southeast and northeast sections and

5  multiplied by the freshwater relative to the reference salinity $S_{ref} = 34.95$. The mean freshwater flux due to the eddy fluxes is 0.2 mSv. This is an order of magnitude lower than the freshwater fluxes due to Ekman transport. Repeating this calculation for the upper 100 m (a more common choice of the surface layer in the Labrador Sea, e.g. Straneo (2001); Schmidt and Send (2007); Schulze et al. (2016)) we find that the combined freshwater transport to the basin due to Ekman and eddy fluxes is 2.4 mSv. This means that the freshwater flux in the top 30 m makes up 60% of the total freshwater flux over the top 100 m. Of this,

10  more than half is due to Ekman transport. When dividing the freshwater flux of the top 100 m into Ekman transport and eddy fluxes, the Ekman transport alone still accounts for more than 60% of the total 2.4 mSv. Eddy fluxes become more important only when extending the calculation to 200 m.

Two novel results emerge from this study. Firstly, the two seasonally-occurring freshwater pulses identified in the model can be traced to the EGC. The inshore water is the main source of freshening in the basin, seasonally as well as interannually.

This means that Arctic meltwater and runoff from Greenland have the largest influence on the freshwater input to the central Labrador basin. In light of the changing climate, this could mean a reduction in the formation of LSW with the potential for further reduction in the overturning circulation (Robson et al., 2014). Secondly, we show that Ekman transport plays a significant role in the advection of water to the basin. Previous studies concentrated on determining how large a role eddies play in the restratification of the Labrador Sea, but in a region where the freshest waters are concentrated at the surface and

winds are strong, the surface Ekman transport cannot be neglected.





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





**Table 1.** Number of trajectories with different criteria

|  |  | Count | % of total |
|---|---|---|---|
| Total |  | 570,240 |  |
| Crossings <30 m |  | 230,147 | 40% |
|  | Crossing with in 7 mth | 205,929 |  |
|  | • Stay <30 m | 176,790 |  |
|  | • Leave top 30 m | 29,139 |  |
|  |  |  |  |
|  | Crossing later | 24,218 |  |
|  | • Stay <30 m | 20,585 |  |
|  | • Leave top 30 m | 3633 |  |
| Crossings >30 m |  | 1657 | <1% |
| Enter in south |  | 323,084 | 56 % |
|  | • Stay <30 m | 96,926 |  |
|  | • Leave top 30 m | 226,158 |  |
| Stay in basin |  | 15,352 | 3% |
|  | • Stay <30 m | 1453 |  |
|  | • Leave top 30 m | 13,899 |  |





**Table 2.** Correlation of the number of crossings in the southeast/northeast and the EKE and Ekman transport in the same region. The table shows the r-value of each correlation, printed in **bold** if the correlation is significant within 99 % confident levels.

| SOUTHEAST | Ekman | EKE |
|---|---|---|
| Number of crossings | **0.45** | 0.25 |
| Number of inshore crossings | **0.54** | 0.11 |
| Number of offshore crossings | 0.2 | 0.26 |
| NORTHEAST | | |
| Number of crossings | **0.72** | 0.05 |
| Number of inshore crossings | **0.72** | 0.21 |
| Number of offshore crossings | 0.11 | 0.29 |





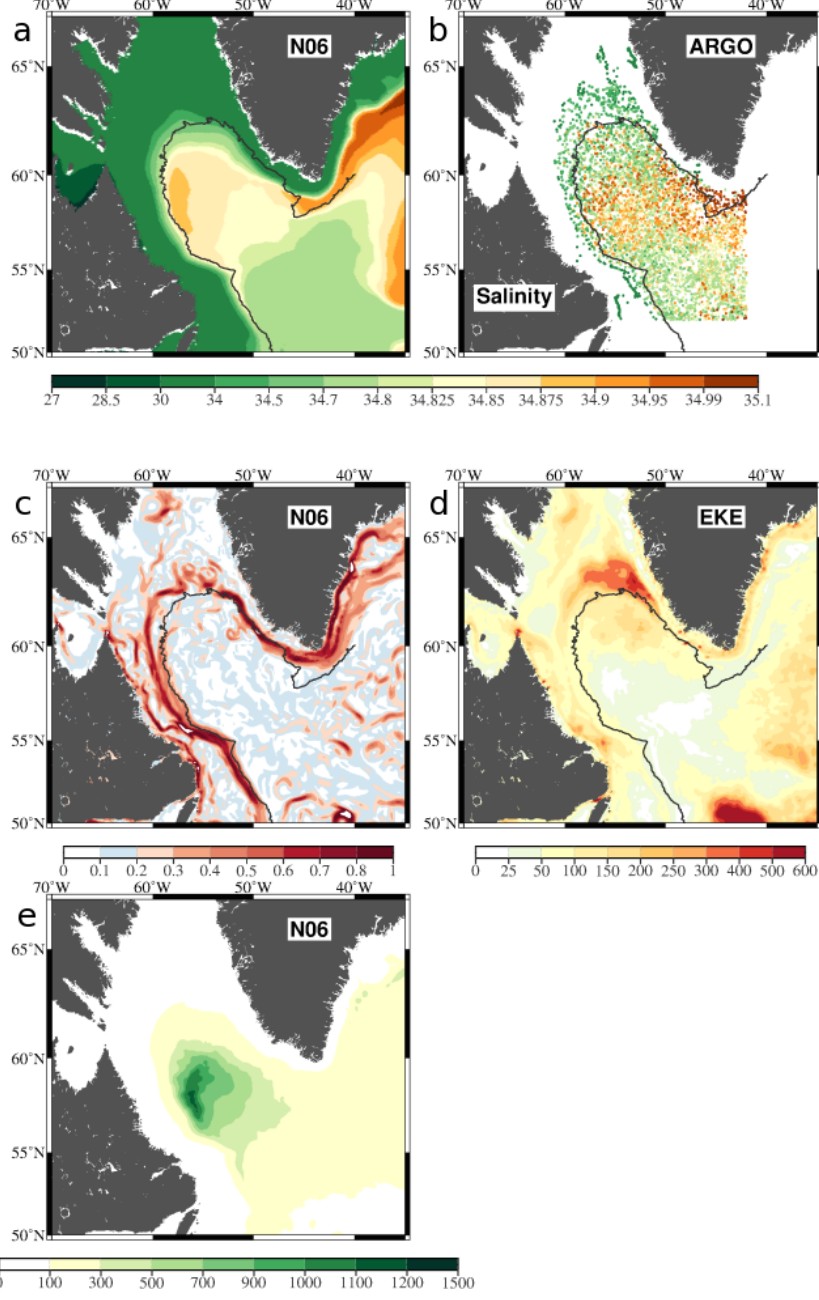

**Figure 1.** a): Mean salinity in the top 100 m (1990 – 2009) from NEMO-N06 b): same as a) but from ARGO for the period 2002 – 2012. c): Speed [cm/s] and d): mean EKE [cm$^2$/s$^2$], (1990 – 2009) derived from the NEMO-N06 model of the top 100 m. e): Mean winter time (Dec – Mar) mixed layer depths [m] from NEMO-N06 for 1990 – 2009.



**Figure 2.** Top: The location of the Labrador Sea (left) and a zoomed in view of the Labrador Sea on the right. The topography is shown in gray contours, spaced in 500 m intervals. The thick contour shows the 2500 m isobath and is referred to as the boundary between shelf and basin in the text. The areas referred to in the study as southeast and northeast are shown in blue and purple, respectively. Red dots denote the release positions of the particles in this study. The five regions referred to as the origin of water are also shown here. The East Greenland Current (EGC) inshore and offshore region are shown as the blue and red box, respectively. Baffin Bay and Hudson Strait are shown as black sections and the North Atlantic region as the yellow line and structures region. Bottom: The number of crossings per origin. East Greenland offshore (red), East Greenland inshore (blue), other regions in the North Atlantic (yellow), unidentified origins (no color), Baffin Bay and Hudson Strait (black).





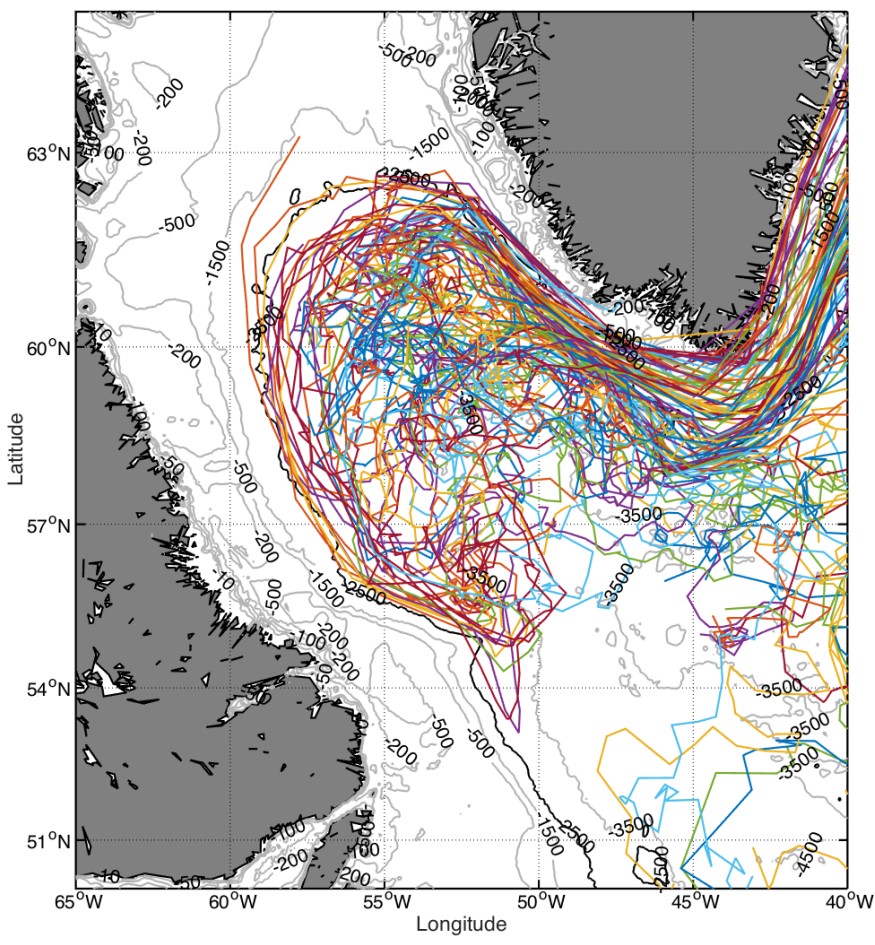

**Figure 3.** Trajectories of 0.01% of the 205,929 trajectories that entered the basin. The trajectories were chosen randomly and are shown in a different color each. Bathymetry is contoured in gray at 500 m intervals with the 2500 m isobaths in black





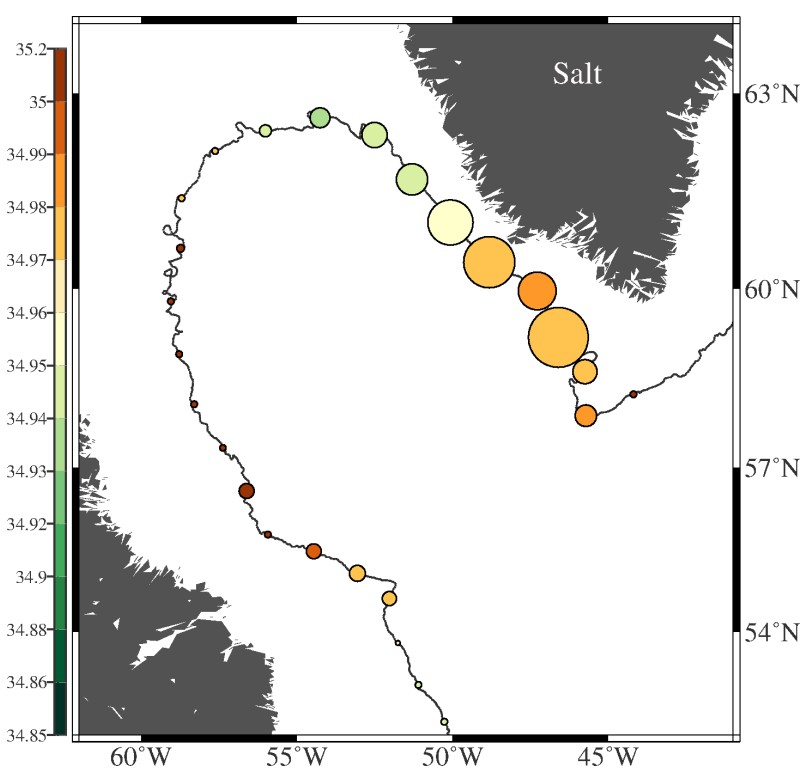

**Figure 4.** The probability of crossings per 100 km along the boundary is indicated by the size of the circles, with larger circles indicating a larger probability. The color shows the mean salinity of the crossings at each section.





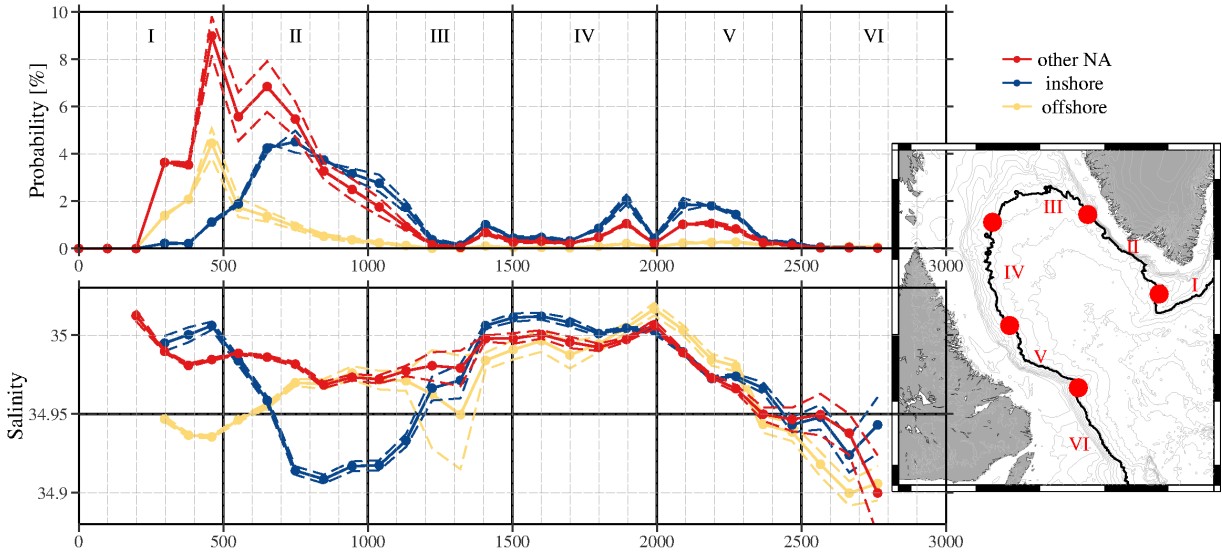

**Figure 5.** a): The probability of crossings per 100 km section (solid line) and the estimated error (dashed line). b): The average salinity of the crossings particles at each 100 km section (solid line) and the associated error (dashed lines). The black horizontal line shows the reference salinity of 34.95 that is used to calculate the freshwater flux. In both panels the vertical lines correspond to the location of the red circles on the map to help orient the reader geographically. Red lines show the EGC offshore water, blue the EGC inshore water and yellow the water from other regions of the North Atlantic.





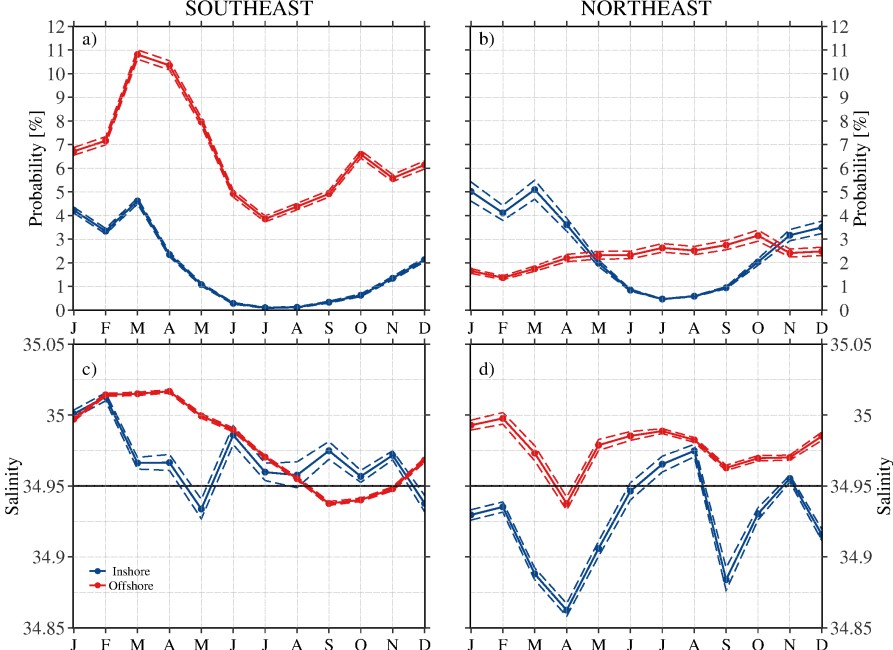

**Figure 6.** a) Seasonal cycle of the probability of particles entering the basin in the southeast and b) northeast, (see Figure 2 for the location of the regions). Seasonal cycle of salinity for particles crossing in the c): southeast and d) northeast. In all panels, the colors show the sources of the water: Blue lines shows water from the EGC inshore region and red the water from the EGC offshore region. The dashed lines show the associated errors.





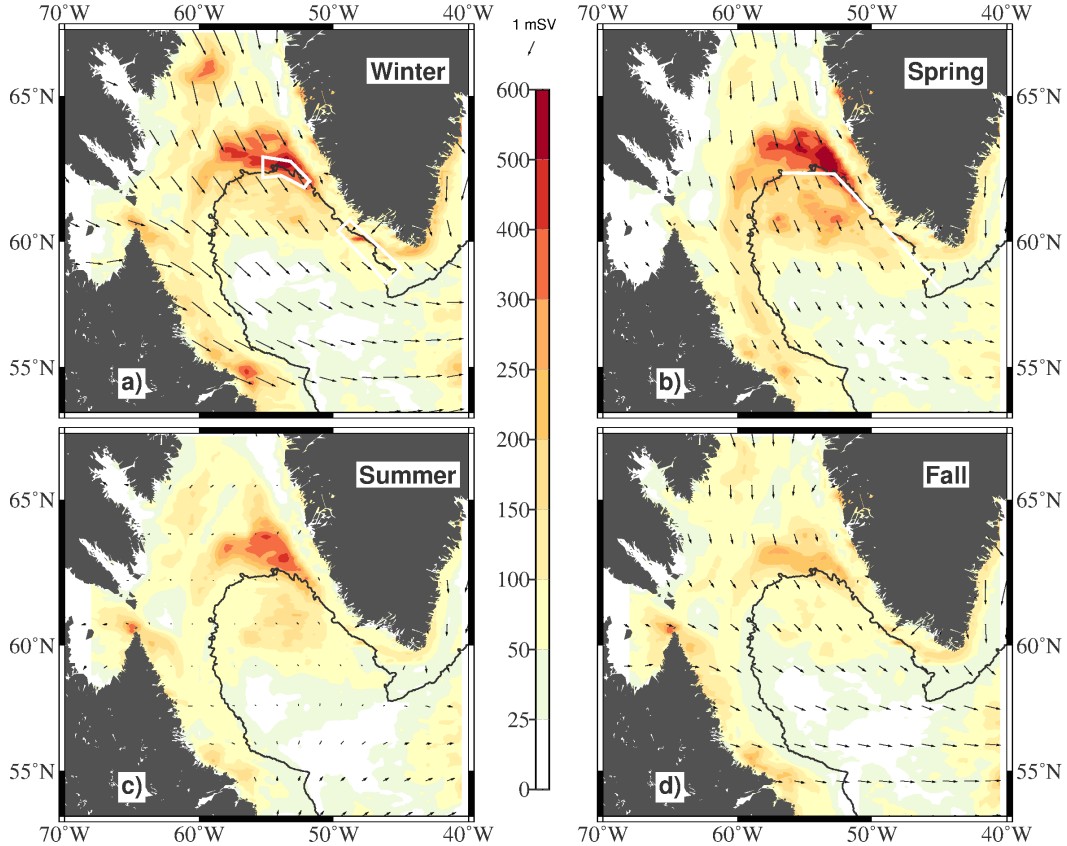

**Figure 7.** Three monthly mean of eddy kinetic energy (color [cm$^2$/s$^2$]) and wind (vectors [m/s]) in the Labrador Sea, 1990 – 2009, for a), Dec – Feb), b), Mar – May), c), Jun – Aug), and d), Sep – Nov). The white boxes in a) show the regions over which EKE is averaged in **Figure 8**. The white lines in b) show the sections across which Ekman transport is calculated.





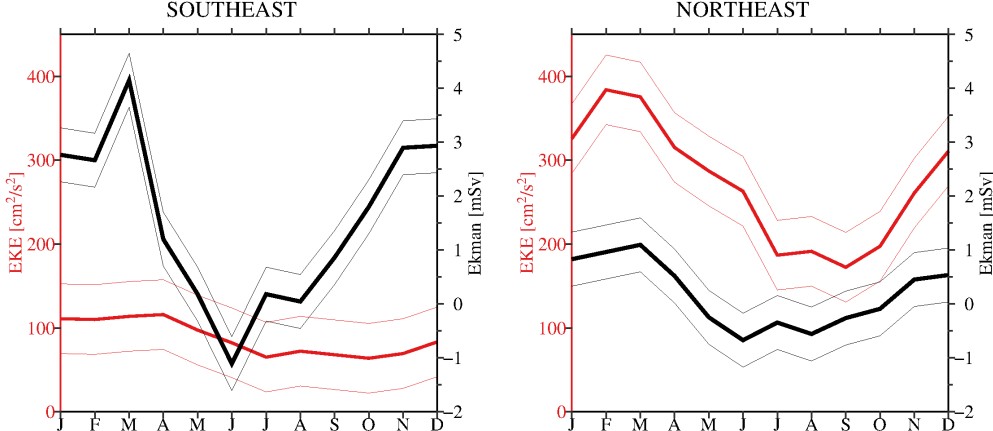

**Figure 8.** Left: The seasonal cycle of EKE (red line) and Ekman transport (black line) (1990 – 2009) in the southeast (See white box and section in **Figure 7**). The thin lines show the associated standard deviation. Right: Same but for the northeast.


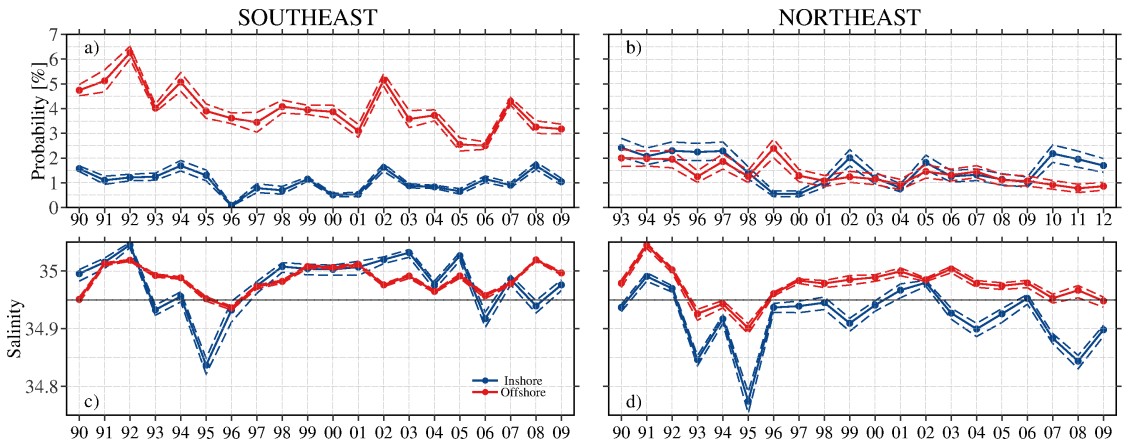

**Figure 9.** The probability of water entering the basin in the a): northeast and b): southeast. The salinities of particles crossing in c): the northeast and d): the southeast. The colors refer to the water's origin: blue shows the EGC inshore water, red the EGC offshore water. The dotted lines show the estimated errors.





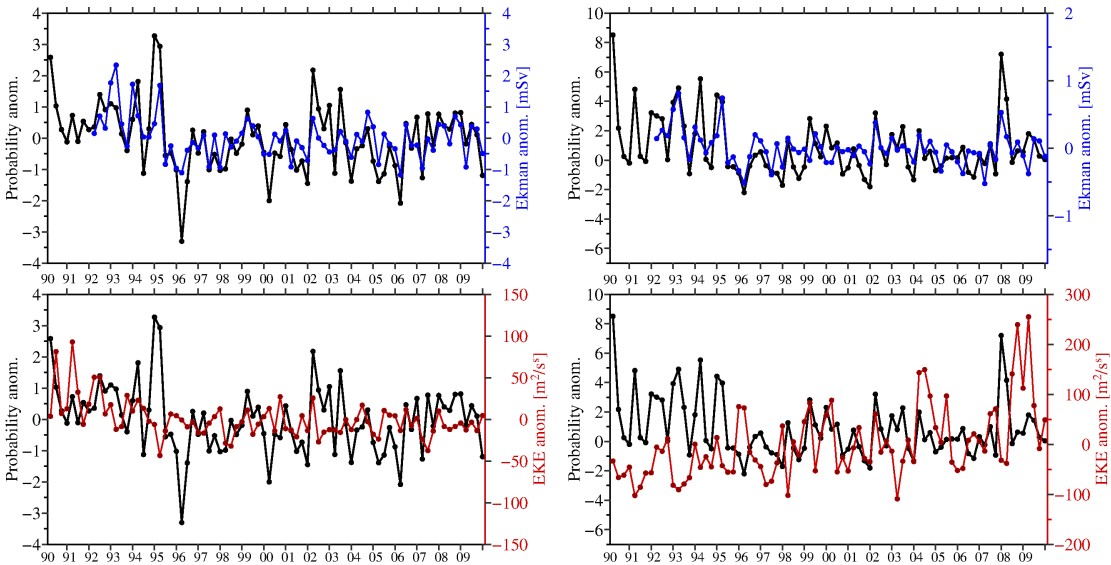

**Figure 10.** Top panels: Three-monthly anomaly of the crossing probability in the southeast (left) and northeast (right), (black lines) and the Ekman transport anomaly in the same regions (blue). Bottom panels: Same as above but for the crossing anomaly (black lines) and EKE anomaly (red).





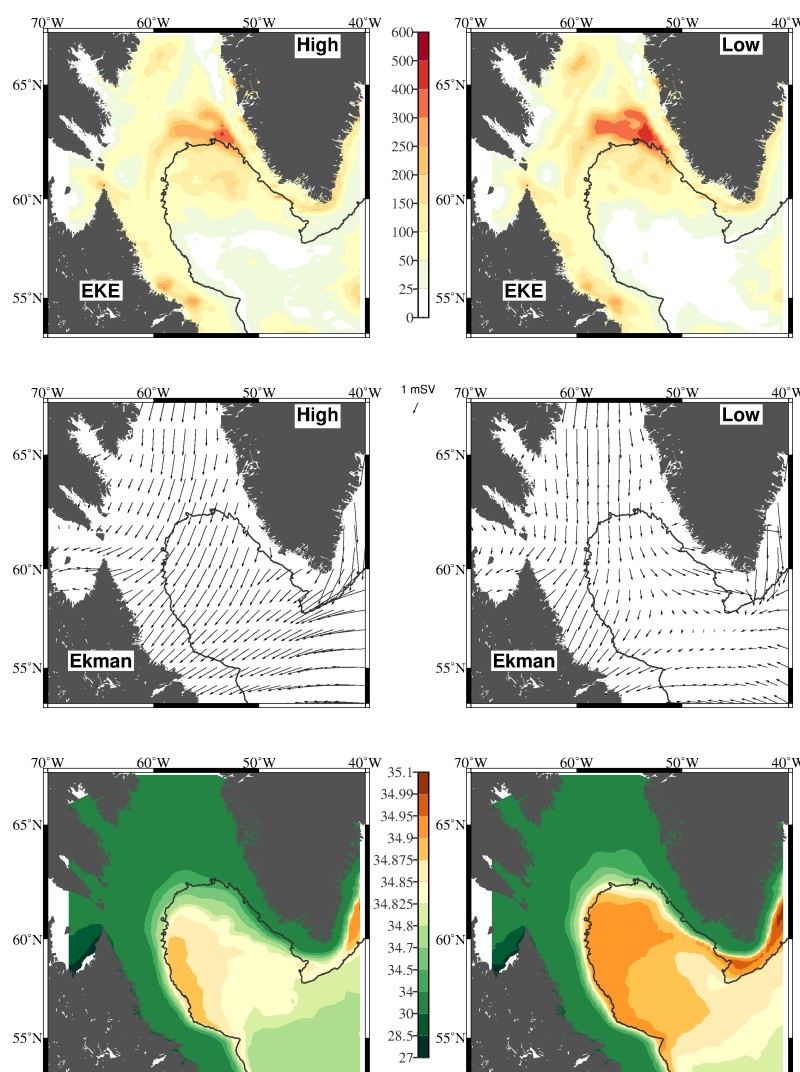

**Figure 11.** Top: The mean surface EKE [cm$^2$/s$^2$] during months with anomalously high (left) and low (right) number of crossings. Middle: Same as the top row but for the Ekman transport, Bottom: Same as top but for the model salinities of the top 30 m.