# Peer review of "Wind-driven transport of fresh shelf water into the upper 30 m of the Labrador Sea"

_Ocean Science, 2018_

## Referee Comment (RC1) · Anonymous Referee #1 · 4 Apr 2018

This manuscript looks at the exchange of fresh shelf water into the Labrador Sea using a high resolution numerical model and Lagrangian trajectories. The authors find that much of the freshwater that reaches the interior of the Labrador Sea comes from the West Greenland Current (which isn't a new result). But they expand on this work, by showing two seasonal pulses, associated with different geographical positions (southeast, northwest) and different salinity waters. Where this work truly expands upon previous studies is showing the key role of wind-driven Ekman transport compared to the typical view of eddy driven exchange.

Given this is an important topic (fate of enhanced high latitude freshwater on water formation in the Labrador Sea), this work is appropriate for the journal. It is a well written paper, easy to follow and understand. Thus it is definitely will be eventually

suitable for publishing in Ocean Science. However, there are a few places where the manuscript could be improved upon. Thus I recommend minor revisions. Details of my comments are given below.

Introduction General: Although the introduction provides a good summary, it feels a bit short. More discussion of previous work related to offshore exchange in the Labrador Sea can be added. Both with respect to observational studies, but especially with respect to previous modelling works. Since the authors are going to dispute the commonly held paradigm that eddies are the main exchange mechanism from the WGC, they need to discuss the previous modelling works that have highlighted that mechanism (and then in the discussion try to bring out why the present results are different). Beyond papers listed such as Chanut et al, there are newer studies such as McGeehan and Maslowski, Gelderoos et al., Kawaski and Hasumi, Saenko et al., Dukhovskoy et al to name a few.

Page 2, Line 11: ? in the references needs to be filled in

Page 2, Line 28: But doesn't the Cooke paper use a very coarse resolution model, making it easy for freshwater to leave the Labrador Current. If so, this point could be clarified

General: At some places in the manuscript the authors report salinities as dimensionless, and in others places use psu as a unit. At the very least the authors must be consistent.

Page 4, Line 19: More detail on the lateral boundary conditions in the region, and the impact of that choice would be useful.

Page 4, Line 31 – is used...

Page 5, 1st paragraph: Changes implemented in the model are listed as 1), 2) and 4). Where is number 3?

Page 5 – in terms of evaluation, given the importance of the West Greenland Current

to the paper, it might be good to see further evaluation of the model representation of this feature. I.e. Don't just focus on the EKE in terms of observational comparisons.

Page 6, Line 22: Badly worded sentence with place/placed used an extra time

Section 2.6 – The calculation of Ekman transport is discussed here, but the sections for which it is computed are not shown until the white line in figure 7. Be good to show that earlier. Additionally, how close is that line to the actual isobaths in the model? Does the line follow a model grid line?

Page 10, line 12 – looks like there is weak EKE in late summer too.

Section 4.2.1 – Does the 3 month averaging remove eddies and thus the damp the potential importance of this term?

Page 12, Line 20: The statement "in the NEMO model..." is not correct. The authors mean in their configuration of the NEMO model, with the given forcing, they find...

Page 12, Line 32: With respect to the statement about higher resolution being needed, doesn't Chanut et al argue that at least 1/15 degree is needed?

Page 13, line 21: Do any of the years mentioned stand out in terms of freshwater transport, melt from the Greenland ice sheet, very positive NAO, etc.?

Table 1: within is one word; Additionally I don't like the phrasing "Crossing Later" – the authors can be more precise and quantitative.

Figure 1: Why are the observations and model field plotted for different time periods (1990-2009 vs 2002-2012)? Can't the results be subsampled to plot everything over the same time period to allow a fairer comparison? Also for the model mixed layer depth, is it based on the default NEMO threshold method? If so, Courtois et al, 2017 show this approach significantly overestimates the actual model mixed layer in deep convection regions.

Figure 4 – Why does it say 'Salt' in the middle of Greenland?

---

## Referee Comment (RC2) · Anonymous Referee #2 · 9 Apr 2018

**Overview**

In this manuscript the authors investigate the sources of freshwater transport in the Labrador Sea, the locations at which freshwater enters the central basin, the dynamical mechanisms responsible for this transport, and the controls on seasonal and decadal variability in the transport. Their tool is an unconstrained 1/12 degree multi-decadal integration of the NEMO coupled ocean/sea ice model, in combination with the offline Lagrangian particle advection tool ARIANE. The authors derive Lagrangian particle back-trajectories for waters in the upper 30m of the central Labrador basin over a 20-year period, and then compute statistics associated with the frequency at which particles cross into the basin and the salinities associated with the crossings.

The authors find that most of the particles originate from the shoreward and offshore

branches of the East Greenland Current (EGC), in agreement with previous studies, and that the particle crossings occur predominantly in what they call the "Northeast" and "Southeast" sectors of the Labrador Sea. The waters entering from the inshore branch are fresher by  $\sim$ 0.1 salinity units on average. The inflowing EGC inshore-sourced water exhibits substantial annual variability in both probability of particle crossings and, in the "Northeast" Labrador Sea, in its salinity. Based on this, the authors infer that inflow of relatively fresh EGC inshore-sourced water occurs in two peaks: one in September, and one around April.

The authors then contrast eddy kinetic energy (EKE, a proxy for eddy particle transport into the basin) and wind-driven Ekman transport as mechanisms underlying the diagnosed particle transport. Both EKE and Ekman transport exhibit seasonal cycles, though the Ekman seasonal cycle is much more pronounced in the "Southeast" section of the Labrador Sea, while EKE is more pronounced in the "Northeast" section of the Labrador Sea, the probability of particles having entered the basin correlates significantly with the wind stress in both the Northeast and Southeast sections, but particularly strongly in the Northeast, where Ekman transport variations explain ~50% of the variance in the particle crossing probability. Based on this, the authors infer that winds control interannual variations in freshwater inflow to the central Labrador basin.

This manuscript addresses an important topic, the analysis is interesting and insightful, and in my opinion this work is worthy of publication in Ocean Sciences. However I have a long list of comments on the manuscript (see below), including some quite strong criticisms of the authors' methodology and the evidence supporting their central conclusions. My most major concerns relate to (i) the authors conclusion that freshwater enters the Labrador basin in two "pulses" each year, which does not seem to be supported by their calculations, and (ii) the authors' decision to focus their particle deployments and particle crossing analyses on the upper 30m of the water column, which inherently biases their results toward wind control of freshwater transport. Therefore, major revisions of the manuscript, likely including substantial additional calculations, will be required to bring this up to a standard appropriate for publication. The manuscript itself is well structured but poorly written: as noted below, there were too many spelling errors, grammatical oddities, and instances of unclear phrasing to list in this review. The manuscript will therefore extensive proof-reading by a native English speaker during revisions.

Comments/questions:

At times I found it difficult to make my way through the manuscript due to the high density of grammatical and spelling errors, and awkward phrasings (in various cases so as to render the meaning unclear). I initially tried to catalogue these errors to pass them on to the authors, but quickly gave up due to the sheer number of them. During revisions the authors should pass the manuscript to a native English speaker for detailed corrections throughout, as I do not consider the current standard of writing to be suitable for publication. Additionally, in other places the writing is rather vague, and I have attempted to identify such instances in comments below.

p1, L10-12; p10, L6-7; p13, L4-5: I am not convinced that the authors' evidence supports this conclusion. I was initially confused by the authors' wording in the abstract, where they claim that they diagnose two peaks of freshwater transport into the LS; I wondered why they distinguished the first peak as being associated with "a large number of shelf water particles". After reading the manuscript, it became clear that the converse statement is more relevant: the second peak in the salinity anomaly (in the particles from the inner EGC entering via the "Northeast" section of the LS) is not associated with a large number of shelf water particles, at least not compared to the first. Given that the actual freshwater flux may be expected to be related to the product of the salinity anomaly with the number of particles, is this second peak even worthy of note? Perhaps the authors could produce some quantitative estimates of the freshwater flux associated with this "peak" to support their conclusion, but my reading of their current results is that there is really only one peak in the freshwater transport into the LS, occurring around April.

p1, L16-21: This discussion should be accompanied by supporting citations.

p1, L19: "the salty basin" - does this simply refer to the central Labrador Sea? In general I found the authors' "basin" terminology to be ambiguous. They should clarify how they and previous authors distinguish basin from shelf, and ensure that nomenclature is consistent with previous studies.

p2, L11: There appears to be a missing citation here (replaced instead with a "?").

p2, L23-24: Do the authors' findings not contradict this? By my reading, the authors diagnose a much stronger Spring pulse of freshwater than in Fall. In the Discussion (p13, L7-8) the authors explicitly state that the opposite is true, and that their findings are consistent with Schmidt and Send 2007. I think a more candid discussion of differences between the authors' findings and previous results is required, as currently this is difficult to reconcile.

p3, L8; p4, L33; p5, L20; p9, L12; p13, L13 (and more; I gave up listing them): At various points the authors make vague statements such as "substantial buoyancy is lost", "the model well represents", or "a strong WGC". Without some quantitative measure, descriptions like "substantial", "well" and "strong" become simply subjective judgements on the part of the authors.

p4, L5-6: Please check the value given for the bi-Laplacian viscosity. If this value were used, the time scale for viscous mixing at the grid scale (4km) would be on the order of 10,000 years!

p4, L9: Is "integrated" the correct word here. If I understand correctly, DRAKKAR is a reference surface forcing dataset with components drawn from various existing datasets, rather than a model that is integrated forward in time.

p4, L24: Please state the data source used for the river runoff.

p4, L26: In addition to bottom friction, pressure forces also exchange momentum between the ocean and the solid earth. p5, L2-4: The authors appear to have omitted item 3) from their list of 4 changes to the NEMO model. Also, what changes were made to the (presumably sea floor) topography?

p5, L9-11: I disagree with this statement. The correct location and magnitude of the ML depths shows that NEMO accurately represents the ML depths. It is a point in favor of NEMO accurately representing the LS state and circulation in general, but is hardly a clear-cut demonstration of the model fidelity.

p5, L11-12: Is this statement based on model experiments, or is it simply a speculation?

p5, L19: The model and ARGO salinity distributions look qualitatively different to me: there are many ARGO profiles measuring relatively low salinity in the middle of the LS basin, and the shape of the high-salinity region looks to be quite different. Perhaps this is simply due to my subjective interpretation of Fig. 1. To remove the ambiguity here, the authors could provide quantitative metrics of the similarity between the modeled and Argo-derived salinities. Perhaps some of the apparent disagreement stems from the seasonal cycle in the measurements? The authors hint at this on L24. but do not show any data on the model vs. Argo differences in the seasonal cycle.

p5, L26: "in many studies" is not a suitable substitute for citations

p6, L9: Where is "outside" the 2500m isobath? Toward greater depths or toward shallower depths?

p6, L14-15: This statement should be supported by evidence if the authors plan to retain it in the manuscript.

p6, L29-30: At various points the authors' descriptions of the particles becomes confused by the fact that they are calculating back-trajectories, so e.g. it is difficult to tell what "the last time" a particle crosses the LS boundary actually means. In this example the ambiguity is between the first chronological crossing and the first crossing that

occurs during backward time-integration.

p7, L2-3: This is an important methodological point that requires more explanation, and in fact I am concerned that this choice biases the author's results toward wind control of particle crossings. The authors only deploy particles within the top 30m, (approximately within the Ekman layer) and only count particles as having "crossed" into the LS central basin if they do so within the top 30m. On p6, L22 the authors claim that "most freshwater is contained in the upper 30m". First, how much is "most"? Second, storage depth does not necessarily equate to transport depth - it is quite plausible that freshwater could enter over a greater range of depths, but only accumulate in the upper 30m. If the authors had deployed their particles over a greater depth range then they could defend their focus on the upper 30m, as they could compare freshwater inflow in the upper 30m against that occurring deeper than 30m. I consider this to be guite a serious caveat: this choice could potentially explain the apparent dominance of Ekman transport over eddies in controlling the diagnosed interannual variability in freshwater transport into the central LS, and the discrepancy between the relative magnitudes of authors' diagnosed "pulses" of freshwater inflow and those reported in previous studies.

p7, L11-12: I am confused by this statement: don't the authors define "entering the basin" to mean that particles have crossed the 2500m isobath? Perhaps this relates to my earlier comment about the authors' vagueness in referring to "the basin".

p7, L19-23: The criteria listed here are not mutually exclusive: do any particles satisfy multiple criteria? If so, is the determination of their origin performed following the logic indicated in these sentences?

p7, L30-31: Difficult to parse because "end of their lifetime" actually refers to the chronological starting position of the particles - see earlier comment on the clarity of the authors' description of the particle trajectories.

p8, L24-25: I found the authors' geographical descriptions confusing because "southeast" actually refers to the eastern side of the LS region in which particles are deployed, while "northeast" actually refers to the northern tip of this region. I suspect other readers might similarly be misled by this terminology, and recommend changing to something more intuitive.

p9, L24-31 (but also at various other points in the manuscript): The authors mischaracterize the probabilities that the calculate as e.g. the "probability of particles ... to enter the basin" (note that here the grammatical oddities are the authors'). The authors calculate the probability of particles having originated from a given region, given that their back-trajectories crossed the LS perimeter. This is different from the probability of waters originating in, e.g., the EGC inshore region crossing into the central LS to calculate this the authors would need to compute forward trajectories for particles initialized throughout the EGC inshore region. Strictly speaking, the probability that the authors' particles enter the basin is 100% because their trajectories all end in the central LS. The authors should rewrite all sections of the manuscript that discuss these probabilities to accurately characterize the results. E.g. on p10, L1-2, "inshore water is about twice as likely as offshore water to enter" might be more accurately written as "entering water is twice as likely to have originated from inshore as to have originated from offshore".

p11, L18-19: The authors describe the correlation as "significant", but do not define the criterion for statistical significance.

p13, L30-32: Here the authors explicitly decline to address the mechanism via which EGC offshore water is transported into the basin. I do not think this is acceptable in a manuscript that explicitly aims to quantify the relative roles of different mechanisms of freshwater transport into the LS. This point should be addressed in detail in a revised manuscript.

p14, L4-5: This calculation is likely to be sensitive to the choice of the reference salinity, and may be producing a misleading estimate of the Ekman freshwater flux. The

authors calculate the mean and eddy components of the freshwater flux across the "northeast" and "southeast" sections of the LS boundary - a useful complement to the Lagrangian analysis that serves as the focus of the paper. That is they integrate the boundary-normal components of <S-Sref> and  along the boundary, where angle frackers < > denotes a time average. Now, the eddy component is insensitive to Sref because  $\langle u' \rangle = \langle S' \rangle = 0$  by definition, so  $\langle u'(S-Sref)' \rangle = \langle u'S' \rangle + \langle u'Sref \rangle =$ Sref = . However, the mean component is <S-Sref> = <S>- <Sref>. If the boundary integral of the boundary-normal component of  is non-zero (which seems very probable given the short lengths of the "northeast" and "southeast" boundary segments, and the prevailing northwesterly winds), then changing Sref will change the computed freshwater flux. Given that the choice of Sref is arbitrary, this renders the authors' estimate of the Ekman freshwater flux arbitrary. A solution is to integrate both the eddy and mean components over the full ocean depth, and to perform the integral along a contour of the time-mean depth-integrated streamfunction - this guarantees that the along-contour integral of  is zero, and therefore removes the arbitrariness introduced by Sref.

p14, L6: The authors equate the mean freshwater transport with the Ekman transport, but the mean flow need not be entirely Ekman - are the authors sure that other contributions to the cross-boundary mean flow are small?

p14, L9-10: I think this sentence is a reasonable take-home message from the study, in contrast to the abstract, which I suspect rather over-states the strength of the authors' conclusions (see other comments above on the methodology).

Fig. 2: How did the authors select this particular pattern of particle deployment? I am struggling to discern the rationale behind the particular pattern shown here.

Fig. 4: I initially thought that the authors had chosen to rename "Greenland" as "Salt", before realizing their intent. Perhaps they could move this label to the left of the figure?

Fig. 4: Please provide a scale for the probabilities associated with the sizes of the

circles.

Fig. 6: A legend would improve the clarity of this figure.

Fig. 8: The authors use EKE as a proxy for the freshwater transport by eddies in their consideration of seasonal and interannual variability. However, EKE alone does not dictate the eddy transport - a better proxy would be something like the square root of EKE multiplied by the salinity difference across the LS boundary. How much seasonal/interannual variability is there in this gradient?

Fig. 10: This figure does not distinguish between waters originating from the EGC inshore and EGC offshore regions. Given that it appears to be the EGC inshore waters that are primarily responsible for the freshwater transport, it would be prudent to make this distinction, particularly given the potential impact on the correlation between winds/EKE and particle crossings.

Fig. 10: Why does the Ekman transport estimate only go back as far as 1992?

Fig. 10: The authors should highlight the differing axis ranges between the panels, as this might mislead readers - in fact I would argue that the axis ranges should be identical for this reason.

Fig. 10: How strong are the computed correlations if annual, rather than three-month, averages are used? Much of the correlation might simply be due to the strong seasonal cycles present in the time series.

Fig. 10: Plotting the probability anomaly over time may actually produce misleading results, because this only measures the number of particle crossings relative to the numbers of crossings in other sections of the LS perimeter. That is, a probability anomaly could arise due to more/fewer particles crossing the northeast section, or it could arise due to fewer/more particles crossing elsewhere. I would recommend switching to a measure of the absolute number of particles crossing to remove this ambiguity.

---

## Author Comment (AC1) · 29 May 2018

**Response to reviewer 1**

May 7, 2018

The authors thank the reviewer for their careful reading of our discussion paper, and for their helpful and constructive comments regarding its content and improvement. The text of the review is reproduced below in black type; our comments are in blue; and changes to the original discussion paper are presented in italics.

This manuscript looks at the exchange of fresh shelf water into the Labrador Sea using a high resolution numerical model and Lagrangian trajectories. The authors find that much of the freshwater that reaches the interior of the Labrador Sea comes from the West Greenland Current (which isn't a new result). But they expand on this work, by showing two seasonal pulses, associated with different geographical positions (southeast, northwest) and different salinity waters. Where this work truly expands upon previous studies is showing the key role of wind-driven Ekman transport compared to the typical view of eddy driven exchange. Given this is an important topic (fate of enhanced high latitude freshwater on water formation in the Labrador Sea), this work is appropriate for the journal. It is a well written paper, easy to follow and understand. Thus it is definitely will be eventually suitable for publishing in Ocean Science. However, there are a few places where the manuscript could be improved upon. Thus I recommend minor revisions. Details of my comments are given below.

**Introduction General:** Although the introduction provides a good summary, it feels a bit short. More discussion of previous work related to offshore exchange in the Labrador Sea can be added. Both with respect to observational studies, but especially with respect to previous modelling works. Since the authors are going to dispute the commonly held paradigm that eddies are the main exchange mechanism from the WGC, they need to discuss the previous modelling works that have highlighted that mechanism (and then in the discussion try to bring out why the present results are different). Beyond papers listed such as Chanut et al, there are newer studies such as McGeehan and Maslowski, Gelderoos et al., Kawaski and Hasumi, Saenko et al., Dukhovskoy et al to name a few.

Thank you for the suggested papers. We have highlighted previous model (in addition to Chanut et al.) that suggest that eddies are the main exchange mechanism from the WGC. Additional discussion was added.

Page 2, Line 11: ? in the references needs to be filled in
This has been fixed

Page 2, Line 28: But doesn't the Cooke paper use a very coarse resolution model, making it easy for freshwater to leave the Labrador Current. If so, this point could be clarified
Yes, Cooke's paper uses a ¼ degree model. We now also noted this in the manuscript.

> **P3. L80.** *Using a $1/4^o$ model, Cooke et al. (2014) argue that the instabilities could indicate a direct connection between the Labrador Current and central basin salinities. Such a connection would further support the idea of a Labrador Current source to the fall freshening in the central Labrador Sea, but the dynamics are not further discussed and the coarse model allows freshwater to leave the Labrador Current more easily than might be the case in the real ocean.*

**General:** At some places in the manuscript the authors report salinities as dimensionless, and in others places use psu as a unit. At the very least the authors must be consistent.

Thank you for noting this. We made sure that this is consistent throughout the manuscript opting for the more modern dimensionless salinity.

Page 4, Line 19: More detail on the lateral boundary conditions in the region, and the impact of that choice would be useful.

We limit the information here to the sentence:

> *P6. L.178. " No-slip conditions are implemented at the lateral boundaries - except in the Labrador Sea where a region of partial slip is applied. This is done to favor the break up of the West Greenland Current into eddies (as observations have suggested)."*

Page 4, Line 31 – is used…

This has been changed as suggested.

Page 5, 1st paragraph: Changes implemented in the model are listed as 1), 2) and 4). Where is number 3?

This was a typo and has been fixed. We also re-worded this paragraph slightly

> *P7. L.194. To improve the NEMO 1/4$^o$ run, changes were incorporated in the 1/12$^o$ run used here to better represent   boundary currents, interannual variability and depth of mixed layers. These changes were: 1) more consistent wind forcing reaching back to 1958 (more information at www.drakkar-ocean.eu/forcing-the-ocean/the-making-of-the-drakkar-forcing-set-dfs5), 2) steeper topography along the Greenland Coast and 3) use of a partial slip along western Greenland. Together with the changes in topography, the partial slip condition promotes the formation of eddies in this region which results in improved salinity and velocities fields (Figure 1). The simulation used in this study was previously used in other studies of the North Atlantic, one of which found that the model represents the variability of heat transport at 26.5$^o$ N.*

Page 5 – in terms of evaluation, given the importance of the West Greenland Current to the paper, it might be good to see further evaluation of the model representation of this feature. I.e. Don't just focus on the EKE in terms of observational comparisons.

Evaluating the West Greenland Current in the model would be useful to understand if the transport and freshwater content of the WGC in the model agrees with observations. Here we decided to concentrate on the EKE since this is regarded as a measure of the West Greenland Currents stability and the region from which eddies are most commonly shed.

Page 6, Line 22: Badly worded sentence with place/placed used an extra time

This sentence has been fixed

> *P.10 l.286. To determine the impact of wind vs. eddies on surface freshwater fluxes into the Labrador Sea, we release particles at three different depths (0 m, 15 m, and 30 m).*

Section 2.6 – The calculation of Ekman transport is discussed here, but the sections for which it is computed are not shown until the white line in figure 7. Be good to show that earlier.

Additionally, how close is that line to the actual isobaths in the model? Does the line follow a model grid line?
We added the sections (shown in Figure 7) to Figure 2.
The sections do not follow a model grid line. Instead it they smooth the isobaths to create a straight line. However, this was tried with multiple lengths of sections (not shown) and we conclude that changing the angle and/or length of the sections does not change the overall results.

Page 10, line 12 – looks like there is weak EKE in late summer too.
We have re-worded this to be more quantitative rather than to refer to the EKE as "strong/weak"

> *p.15 l.449 Three-monthly composites of EKE and wind speeds show that the northeast portion of the Labrador Sea experiences EKE of up to 500 $cm^2/s^2$ in the spring and winter, up to 400 cm2/s2 in the summer and up to 200 $cm^2/s^2$ in the fall.*

Section 4.2.1 – Does the 3 month averaging remove eddies and thus the damp the potential importance of this term?
This is a very interesting point. Averaging SSH in time would remove some eddy effects. However, here we have calculated EKE prior to averaging, meaning that periods of strong eddy activity will still have a large value in the 3 month averaging used. In addition, EKE does not only dictate eddy transport, but also indicate variability of the boundary current. When it is large/eddying, it is expected to result in the formation of eddies. For both these reasons, we believe that averaging in this case will not dampen the potential importance of this term.

Page 12, Line 20: The statement "in the NEMO model…" is not correct. The authors mean in their configuration of the NEMO model, with the given forcing, they find…
Changed as recommended.

Page 12, Line 32: With respect to the statement about higher resolution being needed, doesn't Chanut et al argue that at least 1/15 degree is needed?
Chanut et al use a 1/15 degree model and argue that it performs better than the 1/3 degree model. They do not compare their result to lower (i.e. 1/12 degree) or higher resolution models.

Page 13, line 21: Do any of the years mentioned stand out in terms of freshwater transport, melt from the Greenland ice sheet, very positive NAO, etc.?
We looked into this and nothing really stands out in terms of freshwater, runoff and NAO. The only relationship that might be important is the deep convection that was observed in 2007 – 2008. As for the other years, we are not sure what caused the presence of fresher water. It would be interesting to look at this closer in the model. Maybe a composite of these years, or an analysis targeted to these years versus the other years would help understand this question better.

> *p.20 l.592 Our results show that water entering the Labrador Sea basin was freshest in in the mid-1990s, with other maxima in 1999, the early 2000s and mid-2000. The freshening in the mid-1990s is likely to be related to the freshening observed by Häkkinen (1999), with the freshest waters located on the shelves. Several other years stand out as well, such as 1999, 2003 -- 2004 and 2007 -- 2008. The water responsible for these freshening periods originates in the inshore part of the EGC. A surface freshening signal in 2007 -- 2008 was found in observations, as well as the model. This is also the year during which deep convection was observed again after a long period of absence (Våge et al.*

*2008). It is not clear what exactly caused the freshening periods since the NAO is neither strongly positive nor strongly negative and there is no obvious increase in Greenland runoff at these times.*

Table 1: within is one word; Additionally I don't like the phrasing "Crossing Later" – the authors can be more precise and quantitative.

We have made the term "Crossing Later" more precise, changing it to "Crossing after 7 mth" and the typo has been corrected.

Figure 1: Why are the observations and model field plotted for different time periods (1990-2009 vs 2002-2012)? Can't the results be subsampled to plot everything over the same time period to allow a fairer comparison? Also for the model mixed layer depth, is it based on the default NEMO threshold method? If so, Courtois et al, 2017 show this approach significantly overestimates the actual model mixed layer in deep convection regions.

Comparing observations and model fields for the same time period is a great suggestion and has been done. The mean of the model fields and ARGO data are now calculated for the timeperiod of 2002 – 2009.

Yes, the mixed layers are based on the default NEMO threshold method. Thank you for pointing out the Courtois et al. 2017 paper. We now reference it in the revised manuscript (p.7 l.210).

Figure 4 – Why does it say 'Salt' in the middle of Greenland?

'Salt' was removed from the figure

---

## Author Comment (AC2) · 29 May 2018

**Response to reviewer 2**

May 7, 2018

The authors thank the reviewer for their careful reading of our discussion paper, and for their helpful and constructive comments regarding its content and improvement. The text of the review is reproduced below in black type; our comments are in blue; and changes to the original discussion paper are presented in italics.

**Overview:** In this manuscript the authors investigate the sources of freshwater transport in the Labrador Sea, the locations at which freshwater enters the central basin, the dynamical mechanisms responsible for this transport, and the controls on seasonal and decadal variability in the transport. Their tool is an unconstrained 1/12 degree multi-decadal integration of the NEMO coupled ocean/sea ice model, in combination with the offline Lagrangian particle advection tool ARIANE. The authors derive Lagrangian particle back-trajectories for waters in the upper 30m of the central Labrador basin over a 20-year period, and then compute statistics associated with the frequency at which particles cross into the basin and the salinities associated with the crossings.

The authors find that most of the particles originate from the shoreward and offshore branches of the East Greenland Current (EGC), in agreement with previous studies, and that the particle crossings occur predominantly in what they call the "Northeast" and "Southeast" sectors of the Labrador Sea. The waters entering from the inshore branch are fresher by 0.1 salinity units on average. The inflowing EGC inshore-sourced water exhibits substantial annual variability in both probability of particle crossings and, in the "Northeast" Labrador Sea, in its salinity. Based on this, the authors infer that inflow of relatively fresh EGC inshore-sourced water occurs in two peaks: one in September, and one around April.

The authors then contrast eddy kinetic energy (EKE, a proxy for eddy particle transport into the basin) and wind-driven Ekman transport as mechanisms underlying the diagnosed particle transport. Both EKE and Ekman transport exhibit seasonal cycles, though the Ekman seasonal cycle is much more pronounced in the "Southeast" section of the Labrador Sea, while EKE is more pronounced in the "Northeast" section. On interannual time scales, the probability of particles having entered the basin correlates significantly with the wind stress in both the Northeast and Southeast sections, but particularly strongly in the Northeast, where Ekman transport variations explain 50% of the variance in the particle crossing probability. Based on this, the authors infer that winds control interannual variations in freshwater inflow to the central Labrador basin.
We have based this statement on the quantitative result noted in Table 2. From this, we see that the Ekman transport variations explain more than 70 % of the variance in the particle crossing probability. In addition the paper concludes that wind controls interannual variations in freshwater inflow of the top 30 m to the central Labrador basin.

This manuscript addresses an important topic, the analysis is interesting and insightful, and in my opinion this work is worthy of publication in Ocean Sciences. However I have a long list of comments on the manuscript (see below), including some quite strong criticisms of the authors' methodology and the evidence supporting their central conclusions. My most major concerns relate to (i) the authors conclusion that freshwater enters the Labrador basin in two "pulses" each year, which does not seem to be supported by their calculations, and (ii) the authors' decision to focus their particle deployments and particle crossing analyses on the upper 30m of the water column, which inherently biases their results toward wind control of freshwater transport.

Therefore, major revisions of the manuscript, likely including substantial additional calculations, will be required to bring this up to a standard appropriate for publication. The manuscript itself is well structured but poorly written: as noted below, there were too many spelling errors, grammatical oddities, and instances of unclear phrasing to list in this review. The manuscript will therefore extensive proof-reading by a native English speaker during revisions.

**Comments/questions:**
At times I found it difficult to make my way through the manuscript due to the high density of grammatical and spelling errors, and awkward phrasings (in various cases so as to render the meaning unclear). I initially tried to catalogue these errors to pass them on to the authors, but quickly gave up due to the sheer number of them. During revisions the authors should pass the manuscript to a native English speaker for detailed corrections throughout, as I do not consider the current standard of writing to be suitable for publication. Additionally, in other places the writing is rather vague, and I have attempted to identify such instances in comments below.
We apologize for the errors in the manuscript. We note, however, that the other reviewer called the paper "well-written" and did not have the same comments regarding the language.

This reviewed version of this manuscript was sent to a professional editor for revision and we are positive that grammatical and spelling errors are no longer present.

p1, L10-12; p10, L6-7; p13, L4-5: I am not convinced that the authors' evidence supports this conclusion. I was initially confused by the authors' wording in the abstract, where they claim that they diagnose two peaks of freshwater transport into the LS; I wondered why they distinguished the first peak as being associated with "a large number of shelf water particles". After reading the manuscript, it became clear that the converse statement is more relevant: the second peak in the salinity anomaly (in the particles from the inner EGC entering via the "Northeast" section of the LS) is not associated with a large number of shelf water particles, at least not compared to the first. Given that the actual freshwater flux may be expected to be related to the product of the salinity anomaly with the number of particles, is this second peak even worthy of note?
It is true that the second peak is not associated with a particularly large number of crossings (compared to the first peak) but we do believe that it is still worth noting. Freshwater anomalies can occur because a large amount (large number of particles) of freshwater enters the region, or because a smaller amount of really fresh water enters the reason (e.g. during the second peak). Also not that during the first peak a large amount of salty water enters the basin in the Southeast. This could have the effect of balancing the high number of crossings of freshwater in the northeast. Hence the second peak might even be stronger in terms of how the freshwater impacts the basin, since in the fall the water entering in the southwest is much fresher.
Perhaps the authors could produce some quantitative estimates of the freshwater flux associated with this "peak" to support their conclusion, but my reading of their current results is that there is really only one peak in the freshwater transport into the LS, occurring around April.
We have estimated a freshwater flux from the number of particles that cross into the basin and their salinity (not shown). Unfortunately, the calculation is limited by the model's resolution. One issue is that more than one particle could cross within a Eulerian grid cell but the model would not distinguish this and would instead count the crossing twice. After further consideration, we did not feel that the calculation warranted publication.
Instead we use the probability of fresh/salty water entering the basin. Doing so did not change, but instead confirmed, the correlative findings (between particle crossing probabilities and potential forcing terms) which was also found when initially working with the an estimate of the freshwater flux.

p1, L16-21: This discussion should be accompanied by supporting citations.
Apologies for the omission. References have been added

p1, L19: "the salty basin" - does this simply refer to the central Labrador Sea? In general I found the authors' "basin" terminology to be ambiguous. They should clarify how they and previous authors distinguish basin from shelf, and ensure that nomenclature is consistent with previous studies.
Yes, "the salty basin" does refer to the central Labrador Sea. We define our definition of the basin on p.10, l.293 *"We refer to the Labrador Sea basin as the region that is offshore of the 2500 m isobaths"*. However, we see that it would be useful to the reader to mention this definition sooner, and have added the following in the introduction:
   p.1 l.30 *Offshore of the boundary currents, in the salty basin*, […]

p2, L11: There appears to be a missing citation here (replaced instead with a "?").
This has been fixed.

p2, L23-24: Do the authors' findings not contradict this? By my reading, the authors diagnose a much stronger Spring pulse of freshwater than in Fall. In the Discussion (p13, L7-8) the authors explicitly state that the opposite is true, and that their findings are consistent with Schmidt and Send 2007. I think a more candid discussion of differences between the authors' findings and previous results is required, as currently this is difficult to reconcile.
Our findings indeed support Schmidt and Send's findings. We find a spring pulse, by itself it is stronger than the fall peak, but considering the large number of particles with high salinity that enter the basin at the same time in the model, the overall effect of freshening on the basin is small according to our metrics. The fall peak seems weaker at first glance, but considering that there is relatively fresh water entering in the southeast also, the peak becomes much more significant. We have added an additional comment to clarify this in the *"Seasonality of crossings"* section.

p3, L8; p4, L33; p5, L20; p9, L12; p13, L13 (and more; I gave up listing them): At various points the authors make vague statements such as "substantial buoyancy is lost", "the model well represents", or "a strong WGC". Without some quantitative measure, descriptions like "substantial", "well" and "strong" become simply subjective judgements on the part of the authors.
We have edited the manuscript with an eye on such statements and have reworded them on many occasions.

p4, L5-6: Please check the value given for the bi-Laplacian viscosity. If this value were used, the time scale for viscous mixing at the grid scale (4km) would be on the order of 10,000 years!
Apologies for this typo, it should have read 3 x $10^{11}$ $m^4$/s . This has been corrected.

p4, L9: Is "integrated" the correct word here. If I understand correctly, DRAKKAR is a reference surface forcing dataset with components drawn from various existing datasets, rather than a model that is integrated forward in time.
We can see how "integrated" could be interpreted incorrectly in this context. We have changed the sentence to:
*"It is used for the period 1958 – 2012"*.

p4, L24: Please state the data source used for the river runoff.

Reference has been added.

p4, L26: In addition to bottom friction, pressure forces also exchange momentum between the ocean and the solid earth.
Thank you for pointing that out.

p5, L2-4: The authors appear to have omitted item 3) from their list of 4 changes to the NEMO model. Also, what changes were made to the (presumably sea floor) topography?
The typo of the list numbers has been corrected.
We changed number 2) in the list to "2) steeper topography along the Greenland Coast" to highlight the changes we were referring to.

p5, L9-11: I disagree with this statement. The correct location and magnitude of the ML depths shows that NEMO accurately represents the ML depths. It is a point in favor of NEMO accurately representing the LS state and circulation in general, but is hardly a clear-cut demonstration of the model fidelity.
We agree that the initial discussion overstated the model fidelity based on the measure of ML depths. We have softened the statement to:

> ***p.7 l.207:*** *In the NEMO N06 model, the deepest winter mixed layers in the Labrador Sea basin are located in the western basin, consistent with observations (Pickart et al., 2002; V_age et al., 2008; Schulze et al., 2016), (Figure 1). The model tends to over210 estimate the mixed layers in the Labrador Sea basin (Courtois et al., 2017), but the agreement of the mixed layer depths and location indicates that the boundary current, and advection of freshwater and heat into the basin, are represented well.*
> *Without this representation the basin strati_cation would be weaker and mixing would be stronger. This in turn would result in mixed layers in the wrong location that are much deeper than in the observations. The relationship between fresh shelf water and mixed layers in the basin can be seen in a previous model study (McGeehan and Maslowski, 2011).*

p5, L11-12: Is this statement based on model experiments, or is it simply a speculation?
It is based on theory and a comparison with the previous version of the NEMO model (not shown) that did not have realistic mixed layer depths.

p5, L19: The model and ARGO salinity distributions look qualitatively different to me: there are many ARGO profiles measuring relatively low salinity in the middle of the LS basin, and the shape of the high-salinity region looks to be quite different. Perhaps this is simply due to my subjective interpretation of Fig. 1. To remove the ambiguity here, the authors could provide quantitative metrics of the similarity between the modeled and Argo-derived salinities. Perhaps some of the apparent disagreement stems from the seasonal cycle in the measurements? The authors hint at this on L24. but do not show any data on the model vs. Argo differences in the seasonal cycle.
It is true that there are some differences in the ARGO and model data. However, there are also similarities, such as the general distribution of salty and freshwater in the basin and the magnitude and amplitude of the seasonal cycle.  While we do not show the seasonal cycle, it is described:

> ***P. 8, L 236:*** *"Seasonal cycles of the basin-averaged salinities in NEMO and from Argo data are in phase with peak salinities in February - March and the freshest water in September. Modeled salinities are overestimated by 0.1 between November - June. "*

p5, L26: "in many studies" is not a suitable substitute for citations
We have edited the manuscript with an eye on such statements and have reworded them on many occasions.

p6, L9: Where is "outside" the 2500m isobath? Toward greater depths or toward shallower depths?
"outside" has been changed to "inshore"

p6, L14-15: This statement should be supported by evidence if the authors plan to retain it in the manuscript.
We have referenced Figure 7 to support this statement. Figure 7 shows the seasonal composites of the EKE.

p6, L29-30: At various points the authors' descriptions of the particles becomes confused by the fact that they are calculating back-trajectories, so e.g. it is difficult to tell what "the last time" a particle crosses the LS boundary actually means. In this example the ambiguity is between the first chronological crossing and the first crossing that occurs during backward time-integration.
We agree that this can be confusing, but have made sure that the entire manuscript is consistent in how the direction of the trajectories are described. We have also changed the paragraph referred to here to:

> **p.9 l.295:** *While the particles were released in the basin and tracked backwards, we will refer to there trajectories forward in time (e.g. particles enter the basin and end up at their release point). A particle is considered to have entered the basin if it crossed the 2500 m isobath from shallow into deeper water within the top 30 m of the water column. If a particle crosses the isobath multiple times, only the last crossing before reaching its release point is considered.*

p7, L2-3: This is an important methodological point that requires more explanation, and in fact I am concerned that this choice biases the author's results toward wind control of particle crossings. The authors only deploy particles within the top 30m, (approximately within the Ekman layer) and only count particles as having "crossed" into the LS central basin if they do so within the top 30m. On p6, L22 the authors claim that "most freshwater is contained in the upper 30m". First, how much is "most"? Second, storage depth does not necessarily equate to transport depth - it is quite plausible that freshwater could enter over a greater range of depths, but only accumulate in the upper 30m.
If the authors had deployed their particles over a greater depth range then they could defend their focus on the upper 30m, as they could compare freshwater inflow in the upper 30m against that occurring deeper than 30m. I consider this to be quite a serious caveat: this choice could potentially explain the apparent dominance of Ekman transport over eddies in controlling the diagnosed interannual variability in freshwater transport into the central LS, and the discrepancy between the relative magnitudes of authors' diagnosed "pulses" of freshwater inflow and those reported in previous studies.
This is a good point. It is true that the method might be slightly bias towards Ekman transport, mainly because particles are only released in the Ekman layer. Because of this, we have addressed this issue in the discussion where we show that the surface 30 m make up 60% of the total freshwater flux over the top 100 m and that eddy fluxes become more important only when extending the calculation to 200 m.

Releasing particles over the entire water column would be crucial if attempting to close the freshwater budget of the Labrador Sea basin. This would be very interesting and it is true that eddies might be the dominant means of advecting freshwater to the basin. However, from ARGO floats and repeat hydrography sections by Yashyaev et al. we do not expect the deeper water to be fresh. Typically, below about 100 m the warm and very salty Irminger water dominates. Hence when trying to describe pathways of freshwater into the basin, we have opted to consider the surface layer, since the deeper water of the boundary current has been shown to be salty. This is a choice made throughout, and it does differ from other choices made to investigate the freshwater transport (models) or freshwater content (observations – e.g. Straneo 2006, Häkkinen 1999). While we agree that this choice highlights the freshwater transport by Ekman transport, this is also a meaningful way to distinguish between the layers in the Labrador Sea. Ekman transport is surface intensified, and while we so not attempt to determine the thickness of the Ekman depth, we expect that the top 30 m will capture the variability of the signal. Eddies would be likely to transport both the surface freshwater and the subsurface warm/salty water (Hatun et al 2007) which could actually decrease their role in freshwater transport into the Labrador Sea.

p7, L11-12: I am confused by this statement: don't the authors define "entering the basin" to mean that particles have crossed the 2500m isobath? Perhaps this relates to my earlier comment about the authors' vagueness in referring to "the basin".
The basin in this case is defined as the region offshore the 2500 m isobaths. This definition does exist earlier in the manuscript, where we define the particles that are considered to have crossed the 2500 m isobaths.
Hence the manuscript states that *"Of the remaining 323,084 trajectories that are not categorized as crossings according to the above criteria […]"* In this category fall particles that enter the basin from the south, hence the North Atlantic but have never been in shallower water.

p7, L19-23: The criteria listed here are not mutually exclusive: do any particles satisfy multiple criteria? If so, is the determination of their origin performed following the logic indicated in these sentences?
Yes, the criteria where chosen such that no particle satisfies multiple criteria.

p7, L30-31: Difficult to parse because "end of their lifetime" actually refers to the chronological starting position of the particles - see earlier comment on the clarity of the authors' description of the particle trajectories.
When referring to "end of their lifetime" in the manuscript, we always refer to the end of their one year runtime. This is consistent with Sections 2.2 and 2.3.

p8, L24-25: I found the authors' geographical descriptions confusing because "south-east" actually refers to the eastern side of the LS region in which particles are deployed, while "northeast" actually refers to the northern tip of this region. I suspect other readers might similarly be misled by this terminology, and recommend changing to something more intuitive.
We believe that the naming of our sections is consistent with their location. The southeast refers to the southern part of the eastern side of the Labrador Sea basin, the northeast refers to the northern part of the eastern side of the Labrador Sea. While we could have named the northeast the 'north' it would have been more difficult to describe the more gradually sloping north to the northwest region in the Labrador Sea. While it would have been simpler if the central axis of the Labrador Sea were meridional, we did consider this choice extensively, and opted for the one used in the manuscript as the most generic.

p9, L24-31 (but also at various other points in the manuscript): The authors mischaracterize the probabilities that the calculate as e.g. the "probability of particles ... to enter the basin" (note that here the grammatical oddities are the authors'). The authors calculate the probability of particles having originated from a given region, given that their back-trajectories crossed the LS perimeter. This is different from the probability of waters originating in, e.g., the EGC inshore region crossing into the central LS - to calculate this the authors would need to compute forward trajectories for particles initialized throughout the EGC inshore region. Strictly speaking, the probability that the authors' particles enter the basin is 100% because their trajectories all end in the central LS. The authors should rewrite all sections of the manuscript that discuss these probabilities to accurately characterize the results. E.g. on p10, L1-2, "inshore water is about twice as likely as offshore water to enter" might be more accurately written as "entering water is twice as likely to have originated from inshore as to have originated from offshore".

We have revisited the manuscript keeping this comment in mind. The reviewer is correct that the probability of particles entering the basin is 100% because all particles were released in the basin. While it is true that all particles end up in the basin, here the probabilities refer to the percentage of those particles that did so crossing through a certain region or in a certain time period. This is described in Section 2.5.

p11, L18-19: The authors describe the correlation as "significant", but do not define the criterion for statistical significance.

The reference for the method with which the correlation was calculated is given in:

*__P.17 l.498:__ The timeseries for EKE and Ekman transport are correlated with the probability anomaly using the Pearson method (Thompson and Emery, 2014).*

p13, L30-32: Here the authors explicitly decline to address the mechanism via which EGC offshore water is transported into the basin. I do not think this is acceptable in a manuscript that explicitly aims to quantify the relative roles of different mechanisms of freshwater transport into the LS. This point should be addressed in detail in a revised manuscript.

Unfortunately, addressing all mechanisms of freshwater transport into the Labrador Sea is beyond the scoop of this paper. Here, we have focused on diagnosing the regions of freshwater transport into the basin, and used the additional eddy and Ekman analysis to add insight to those central results. Eddies are the canonical view, while wind-driven transport became the unexpected major player in the study following the interannual analysis, after we discovered that eddy and Ekman transport could not be distinguished based on the seasonal cycles alone. However, we do agree that this is an intriguing question and that it would be great to study not only the impact of wind, but also the impact of other mechanisms on the freshwater transport. As the reviewer addressed earlier, for this particles should be released throughout the entire water column to avoid a bias towards wind driven exchange between the shelves and basin.

p14, L4-5: This calculation is likely to be sensitive to the choice of the reference salinity, and may be producing a misleading estimate of the Ekman freshwater flux. The authors calculate the mean and eddy components of the freshwater flux across the "northeast" and "southeast" sections of the LS boundary - a useful complement to the Lagrangian analysis that serves as the focus of the paper. That is they integrate the boundary-normal components of $\langle u \rangle \langle S\text{-Sref} \rangle$ and $\langle u'(S\text{-Sref})' \rangle$ along the boundary, where angle frackers $\langle \rangle$ denotes a time average. Now, the eddy component is insensitive to Sref because $\langle u' \rangle = \langle S' \rangle = 0$ by definition, so $\langle u'(S\text{-Sref})' \rangle = \langle u'S' \rangle + \langle u'Sref \rangle = \langle u'S' \rangle - \langle u' \rangle Sref = \langle u'S' \rangle$. However, the mean component is $\langle u \rangle \langle S\text{-Sref} \rangle =$

<S> - <Sref>. If the boundary integral of the boundary-normal component of  is non-zero (which seems very probable given the short lengths of the "northeast" and "southeast" boundary segments, and the prevailing northwesterly winds), then changing Sref will change the computed freshwater flux. Given that the choice of Sref is arbitrary, this renders the authors' estimate of the Ekman freshwater flux arbitrary. A solution is to integrate both the eddy and mean components over the full ocean depth, and to perform the integral along a contour of the time-mean depth-integrated streamfunction - this guarantees that the along-contour integral of  is zero, and therefore removes the arbitrariness introduced by Sref.

We agree that the calculation is sensitive to the choice of reference salinity. However, the choice of reference salinity is not arbitrary but was instead defined as the average salinity in the surface layer of the basin. In this way, it is used to determine whether the particular transport of water has a net freshening or a net salinifying effect. Unfortunately, this was not clear in the writing, for which we apologize. We have added a sentence at the beginning of Section 4.

*p.13 l.415:* *To quantify if water is fresh or salty we will refer to a reference salinity of 34.95 - the  average salinity of the top 30 m of the basin between 1990 -- 2009.*

p14, L6: The authors equate the mean freshwater transport with the Ekman transport, but the mean flow need not be entirely Ekman - are the authors sure that other contributions to the cross-boundary mean flow are small?

Actually, here we do not conclude that the mean freshwater transport is equal to Ekman transport. We find here that the mean freshwater flux due to eddy fluxes is a magnitude smaller than the mean freshwater flux due to Ekman transport, which is the only comparison we made as other mechanisms are beyond the scope of the investigation.

p14, L9-10: I think this sentence is a reasonable take-home message from the study, in contrast to the abstract, which I suspect rather over-states the strength of the authors' conclusions (see other comments above on the methodology).

The abstract has been changed to better  represent our conclusion.

Fig. 2: How did the authors select this particular pattern of particle deployment? I am struggling to discern the rationale behind the particular pattern shown here.

The red dots in Figure 2 show the particle release locations. The locations were chosen to be a regular grid covering the entire central basin while remaining away from the mean boundary currents.

Fig. 4: I initially thought that the authors had chosen to rename "Greenland" as "Salt", before realizing their intent. Perhaps they could move this label to the left of the figure?

Label has been removed

Fig. 4: Please provide a scale for the probabilities associated with the sizes of the circles.

A scale has been added

Fig. 6: A legend would improve the clarity of this figure.

A legend is already part of the figure (panel c), but has been made larger for clarity

Fig. 8: The authors use EKE as a proxy for the freshwater transport by eddies in their consideration of seasonal and interannual variability. However, EKE alone does not dictate the eddy transport - a better proxy would be something like the square root of EKE multiplied by the

salinity difference across the LS boundary. How much seasonal/interannual variability is there in this gradient?

That is a great suggestion. For now we decided to use EKE as a proxy for potential eddy activity. While it does not dictate eddy transport, it does show variability which in turn is a good indicator for shedding of eddies. As with the freshwater calculation initially used (see response to comment on p1, 10-12), we anticipate that a true freshwater calculation based on the offline Lagrangian trajectories would be difficult to defend.

Fig. 10: This figure does not distinguish between waters originating from the EGC inshore and EGC offshore regions. Given that it appears to be the EGC inshore waters that are primarily responsible for the freshwater transport, it would be prudent to make this distinction, particularly given the potential impact on the correlation between winds/EKE and particle crossings.

It is true that this Figure only distinguishes between the water originating from the southeast, and northeast and not between water from inshore or offshore EGC. After much debating, we decided to not add the offshore and onshore water to the figure since it makes the figure really busy and hard to understand. However, Table 2 shows the correlations between the EKE and Ekman transport and the inshore and offshore components in the southeast and northeast.

Fig. 10: Why does the Ekman transport estimate only go back as far as 1992?

This has been fixed.

Fig. 10: The authors should highlight the differing axis ranges between the panels, as this might mislead readers - in fact I would argue that the axis ranges should be identical for this reason.

We have highlighted the different axis ranges in the caption. In the end, we opted for distinct ranges as otherwise it would be difficult to see any variability in the left panel (smaller range), if a reader were interested in the southeast region in particular.

Fig. 10: How strong are the computed correlations if annual, rather than three-month, averages are used? Much of the correlation might simply be due to the strong seasonal cycles present in the time series.

The correlations are still strong when considering only the annual average since, as stated in **p.16 l.496:** *"To consider variations beyond the seasonal cycle, the mean seasonal cycle for 1990 – 2009 is removed and the resulting anomalies are shown in Figure 10 [...]".*

Fig. 10: Plotting the probability anomaly over time may actually produce misleading results, because this only measures the number of particle crossings relative to the numbers of crossings in other sections of the LS perimeter. That is, a probability anomaly could arise due to more/fewer particles crossing the northeast section, or it could arise due to fewer/more particles crossing elsewhere. I would recommend switching to a measure of the absolute number of particles crossing to remove this ambiguity

We have considered this and analyzed the figure using both the absolute number as well as the probability anomalies of crossings. The results remain the same and we decided to show probabilities rather than absolute numbers, since this is a measure used throughout the entire paper.

---

## Author Response (AR2)

**Response to reviewer**

August 13, 2018

The authors thank the reviewer for reviewing our paper, and for the helpful and constructive comments regarding its content and improvement. The text of the review is reproduced below in black type; our comments are in blue; and changes to the original discussion paper are presented in italics.

In this revision the authors have made many improvements to the manuscript, addressed most of my concerns from the previous review and substantially improving the quality of the writing. I have included an additional list of comments and questions below.

My major outstanding concern is that this article, as currently presented, is going to lead other scientists to conclude that the transport of freshwater into the LS is principally wind driven. In fact what the authors actually demonstrate is that the freshwater transport into the top 30m of the LS is principally wind-driven, i.e. that the transport within the Ekman layer is principally wind-driven. The authors calculate that 60% of the freshwater transport in the top 100m is wind-driven, and that eddies "become more important" when the calculation is extended to 200m. Therefore, by the authors' own calculation it seems to be that the most reasonable conclusion is that eddies are still the most important mechanism of freshwater transport into the LS.

We agree that the eddies still play a major role in transporting freshwater into the Labrador Sea, but that the wind becomes a major component when considering the surface layer only. We changed the last sentence in the abstract to reflect that.

Based on the authors' results, we can reasonably draw the following conclusions:
1. A component of the freshwater inflow into the LS is controlled by Ekman transport.
2. This component occurs is largely confined to the upper 30m of the water column.
3. This component is smaller (though the authors have not quantified by how much) than the total freshwater transport into the LS by eddies (over all depths).
This is a crucial distinction that is not adequately conveyed by the title, abstract and body of the manuscript. The authors' failure to make this distinction has already caused confusion - in their first review, reviewer 1 made the the following remark:

"Since the authors are going to dispute the commonly held paradigm that eddies are the main exchange mechanism from the WGC, ..."

Again, by the authors own admission, "eddy fluxes become more important only when extending the calculation to 200m", so clearly they cannot be disputing the paradigm that eddies are the main mechanism of transport into the LS. Yet this is what reviewer 1 took away from this manuscript, and presumably what many other readers would too.

We apologize for the confusion some readers seem to encounter when reading the manuscript. We have changed the title and tried to clarify even more throughout the manuscript.
We have also changed the abstract to include the statement that our manuscript only addresses the upper 30 m (see above).
Additionally, we have clarified this in several places though out the manuscript, e.g. line 119, line 418, line 503 and other occasions.

My major recommendation is that the authors change the title, abstract and body of the manuscript to remove the ambiguity as to their results, i.e. that winds control a relatively small component of the freshwater flux confined to the upper 30m. For example:
1. A suitable title would be "Freshwater fluxes into the upper 30m of the Labrador Sea are dominated by wind transport".

*We changed the title to "Wind-driven transport of fresh shelf water in the upper 30 m of the Labrador Sea" and hope to help clarify possible confusions readers might have.*

2. The authors' statement in the abstract that "60% of the top 100m enters the basin in the top 30m along the eastern side" is accurate but misleading, because they reveal in the manuscript that eddies dominate the freshwater flux over the top 200m.
*We believe that this statement can remain in the abstract since it does not refer to either Ekman transport nor Eddies. It is merely there to show that the surface layer is important when considering freshwater transports into the Labrador Sea Basin. However, in the sentence following this we have stated that the importance of Ekman Transport is only discussed for the top 30 m.*

3. The last sentence of the abstract is also misleading: "the year-to-year variability in the freshwater transport … is dominated by wind-driven Ekman transport, rather than eddies". This is based on the Lagrangian trajectory analysis, and only applies to the top 30m of the water column - there is no reason to think this would still be true if the analysis were extended to 200m.
*This has been fixed.*

4. At the end of the manuscript, summary conclusions are drawn without explicitly acknowledging that they apply only to the top 30m of the water column.
*This has been fixed, and we now refer to the top 30 m throughout the discussion.*

As I said in my previous review, I agree with the authors' final statement of the article, which takes a more modest perspective on the results: "in a region where the freshest water is concentrated at the surface and winds are strong, the surface Ekman transport cannot be neglected". I urge the authors to present the rest of the article with a similar perspective.
*We have done so by revising the manuscript and noting that our results are only valid for the upper 30 m.*

Comments/questions:
- - - - - - - - - - -

NOTE: Page and line numbers refer to the tracked changes version of the article.

Regarding variance in crossing probabilities: in their response the authors state that the "Ekman transport variations explain more than 70% of the variance in the particle crossing probability". Table 2 gives an r-value of 0.72 for this correlation in the northeast, and % of variance explained is r^2 = 51.8%. In the southeast it's more like 25% of variance explained.
*This is correct.*

The "peaks" issue: I still take issue with the authors' claim that there are two distinct "peaks" of freshwater inflow in to the (top 30m of) the LS. It is not sufficient to simply judge by eye from Fig. 6 that two peaks exist - clearly I and the authors have reached different subjective conclusions this way, and an objective method of distinguishing "peaks" is needed.
*We have added a sentence to the text that explains how we identify the two pulses of freshwater (line 444). In addition we moved away from calling them "peaks" as this can be misleading. Instead, we refer to the changes in freshwater described in the manuscript as "pulses".*

The authors note that a peak can occur due to a small amount of very fresh water entering the basin, rather than a large volume of somewhat fresh water. Qualitatively, I follow this argument, but the freshwater "peak" in September is actually saltier than in March/April, AND is associated with a 5x lower crossing probability (Fig. 5b,d).

We assume that the reviewer is referring to Figure 6 and not 5 (since Figure 5 does not have panels a-d and also does not show any seasonallity)?

The reviewers statement is true, the September peak is associated with lower crossing probabilities (1% vs. 4%) and somewhat saltier water. When considering both the Southeast and Northeast in Figure 6 the salty offshore water in April the southeast will weaken the very fresh water in the northeast, while the fresh offshore water in September in the southeast, will strengthen the freshening in the fall.

The authors note that there is a large influx of salty water in the Southeast during the March/April peak (see also L532-535), but without some kind of quantification of the relative magnitudes of these fluxes it is impossible to draw objective conclusions. For example, is the salt influx in the Southeast so large that we shouldn't even consider the March/April freshwater inflow in the Northeast to be a "peak" any more?

We realize that this is not a very quantatative statement. However, to quantify this we would need to calculate some sort of freshwater flux. As noted before we have estimated a freshwater flux from the number of particles that cross into the basin and their salinity (not shown). Unfortunately, the calculation is limited by the model's resolution. One issue is that more than one particle could cross within a Eulerian grid cell but the model would not distinguish this and would instead count the crossing twice. Due to these complications, we use the probability of fresh/salty water entering the basin and some less quantitative statements that come with this decision. Doing so did not change, but instead confirmed, the correlative findings (between particle crossing probabilities and potential forcing terms) which was also found when initially working with the an estimate of the
freshwater flux.

The authors state that they "have estimated a freshwater flux from the number of particles that cross into the basin", but that they "did not feel the calculation warranted publication". Again, if the authors cannot defend their conclusions quantitatively then I argue that they should not be drawing those conclusions in the first place.

The above statement refers to the calculation of freshwater not the conclusions drawn from the number of crossings, and probabilities of crossings, nor the changes in salinity, correlations between particle crossing probabilities and potential forcing terms.

L224-225: My previous comment about topographic form stress was intended as a correction: friction is not the only process that extracts momentum from the fluid at the sea floor - bottom form stress does too. Also, how is bottom friction represented in this model? Is there a simple quadratic drag law, or does the model actually simulate vertical mixing/viscosity in the bottom boundary layer?

Please refer to the documentation (provided by the references in our manuscript) for these details. It is not in the scope of this work to explain all details of this model that have previously been published and explained by others.

L251: "represents" -> "represent"

This has been changed.

L265: "study" -> "study"

"Studie" has been changed to "study". Thank you

L266-267: I don't understand what "drastically developing" means - could the authors be more specific.

We shortened this to "produced unrealistic deep convection" to avoid more detail in an already long lengthy section.

L275: "high," -> "high:"
This has been changed as suggested.

L293: "in many studies" is not a substitute for citations. I specifically highlighted this omission in my previous review, and am surprised to see that it has not been rectified.
We apologize that this was missing in the last review. We have now included three example references, but are aware that there are many more.

L298: There is a double parenthesis at the start of the citation on this line.
This has been fixed.

L306: Citation to Luo et al. should not be in parentheses.
This has been fixed.

L325: Citations in a sentence should form a comma-separated list, rather than a semicolon-separated list.
This has been changed

L329: "inshore the" -> "inshore of the"
This has been changed.

L468: "north-" -> "northeast"
This has been changed.

L481–482: Is the crossing speed consistent with the Ekman velocity?
The crossing speeds are at the same magnitude as the Ekman velocity.

L551: Are these transports correct? A few mSv seems very low for Ekman transport across such a long section of the basin edge. I would have expected something on the order of a few Sv.
Yes, the transports are correct. We have noted in the text that these transports are for the upper 30 m only.

L571: "water" -> "waters"
This has been changed.

L571: "probability" -> "probabilities"
This has been changed.

L633: "amount" -> "probability"
This has been changed as suggested.

L635-642: It is interesting that the shelves become saltier during times of low freshwater flux, yet the EKE remains approximately constant. A mixing length scaling for an eddy diffusivity of freshwater would suggest a diffusivity that scales with $EKE^{(1/2)}$, and the freshwater flux is diffusivity * freshwater gradient. So during times when the shelves are fresher, we would expect stronger eddy freshwater fluxes into the LS because there's a larger freshwater gradient between the LS and the shelves, but more-or-less constant EKE and thus diffusivity.
Yes, we agree that this is very intriguing.

L695: I am struggling to discern how the authors have concluded that the second pulse is "about three times stronger than the first pulse".
This statement was not very clear and we have changed this in the manuscript to avoid confusion.

L729—731: Is r=0.72 "remarkably high"? I guess with just over 50% of variance explained it is (barely) justified to say that Ekman transport plays the "primary role in the variability of freshwater transport", but the authors may be over-selling it here.

We believe that this correlation is high, but have deleted the "remarkably" to not oversell our conclusion. Note that we specify that Ekman transport plays a primary role in the variability of freshwater transport "near the surface".

L735-736: … in the upper 30m.

This is now included in the sentence.

L755-763: In their response the authors acknowledge that this calculation of the Ekman freshwater flux will be sensitive to the choice of reference salinity. One could contrive choices of this salinity that make the Ekman freshwater transport equal to zero or that make it larger than the total Ekman volume transport. Please provide some measure of this sensitivity, i.e. take a range of reasonable reference salinities and provide the Ekman freshwater flux as a corresponding range.

We agree that this calculation is sensitive to our choice of reference salinity.
However, all calculations and transports are derived from using the same reference. This is a legitimate way of calculating freshwater (of course the reference salinity has to be stated). Using a different reference salinity would change the stated transport of 2.4 mSv but not our overall conclusions, since they are relative to each other. In addition, the calculations have been done with the reference salinity that represents the mean salinity of the basin over the time period considered here. Hence the results present freshwater transports relative to this mean, a legitimate choice.

L762—763: Please quantify the relative importance of eddy and Ekman transports over the top 200m.

This has been included.

L766—768: … in the upper 30m.

This has been added to the sentence.

L768—769: … in the upper 30m.

This has been added to the sentence.

L772–773: … in the upper 30m.

This has been added to the sentence.

[revised manuscript text omitted]

---

## Author Response (AR3)

September 7, 2018

Dear Dr. Hecht,
Thank you for the good news and picking up on the corrections that had to be done. I made the changes you pointed out. I found a couple of instances of the bibliography capitalization that was wrong. I am hoping I did not miss any.
Everything else has been changed according to your review.
Thank you again,

Lena Schulze Chretien